# Maximizing protein production by keeping cells at optimal secretory stress levels using real-time control approaches

Sebastián Sosa-Carrillo [1], Henri Galez [1], Sara Napolitano[1], François Bertaux[1,2,3] & Gregory Batt [1,3] ✉

Optimizing the production of recombinant proteins is a problem of major industrial and pharmaceutical importance. Secretion of the protein by the host cell considerably simplifies downstream purification processes. However, for many proteins, this is also the limiting production step. Current solutions involve extensive engineering of the chassis cell to facilitate protein trafficking and limit protein degradation triggered by excessive secretion-associated stress. Here, we propose instead a regulation-based strategy in which induction is dynamically adjusted to an optimal strength based on the current stress level of the cells. Using a small collection of hard-to-secrete proteins, a bioreactor-based platform with automated cytometry measurements, and a systematic assay to quantify secreted protein levels, we demonstrate that the secretion sweet spot is indicated by the appearance of a subpopulation of cells that accumulate high amounts of proteins, decrease growth, and face significant stress, that is, experience a secretion burnout. In these cells, adaptations capabilities are overwhelmed by a too strong production. Using these notions, we show for a single-chain antibody variable fragment that secretion levels can be improved by 70% by dynamically keeping the cell population at optimal stress levels using real-time closed-loop control.

Bioproduction is a field of major economic importance and is expected to play an important role for the development of a more sustainable industry[1,2]. Bio-manufactured products include a variety of chemicals, such as alcohols, organic acids, fragrances, antibiotics, and a large range of industrial or pharmaceutical proteins, such as enzymes and antibodies[3,4]. Yeasts are widely used for heterologous protein production. They are inexpensive to grow, easy to engineer, and have extended protein secretory capabilities[5]. This last feature is of great importance because secreting the proteins of interest (POI) greatly facilitates downstream processes and product purification. Yet, protein secretion is a complex multi-stage process. A leader peptide acts as a signal for translocation of the protein from the cytosol to the endoplasmic reticulum (ER). There, the protein is folded and undergoes post-translational modifications and quality control prior to being transported to the Golgi apparatus and being finally secreted[6,7]. Bottlenecks may appear at different stages[3,8–11]. Secretory stress triggers the activation of various adaptive mechanisms. The unfolded protein response (UPR) plays a pivotal role by modulating the expression of hundreds of genes[12–14]. Its action is twofold. On the one hand, it increases trafficking capacities by regulating genes related to translocation, folding, protein maturation or secretion[12,15]. On the other hand, it triggers mechanisms that target the accumulated proteins for degradation, such as ER-associated protein degradation (ERAD)[16,17], ER-phagy[18,19], or ER-reflux[20,21]. Significant research works have focused on finding genetic modifications of the chassis cells that increase trafficking capacities or that mitigate protein degradation[3,8–11]. Unfortunately, these genetic modifications are often chassis- and protein-specific, thus, their

[1]Institut Pasteur, Inria, Université Paris Cité, 75015 Paris, France. [2]Lesaffre International, 101 rue de Menin, Marcq-en-Baroeul, France. [3]These authors contributed equally: François Bertaux, Gregory Batt. ✉e-mail: gregory.batt@inria.fr

identification and implementation in yeast necessitate substantial efforts.

In this study, we propose a fundamentally different strategy. We aim at identifying induction sweet spots, that is, the lowest induction level for which protein secretion is maximal, leveraging adaptation mechanisms that are favorable for secretion and avoiding detrimental ones. We demonstrate that in *Saccharomyces cerevisiae*, induction sweet spots are indicated by the appearance of a subpopulation of cells that accumulate high amounts of proteins, decrease growth, and face significant stress, that is, that experience a secretion burnout. In these cells, adaptation capabilities are overwhelmed by a too strong production, as demonstrated by looking at adaptation-deficient strains (*HAC1* knock-out), which presents the same signature phenotype when secreting easy-to-secrete proteins. Above this level, a further increase of the production demand is associated with an increase of protein degradation, leading to similar or lower levels of secreted protein. Lastly, we show that secretion levels can be improved by 70% by dynamically keeping the cell population at optimal stress levels using real-time closed-loop control. To obtain these results, we combine two innovative approaches. First, we use a small collection of engineered yeast strains expressing various hard-to-secrete POIs whose induction levels can be precisely controlled using a light-responsive promoter and whose secretion levels can be quantified in a systematic manner using magnetic immunobeads, together with a fluorescent reporter for their UPR secretory stress. Second, we use an automated turbidostat-based platform with automated cytometry allowing single-cell measurements at a high temporal resolution over extended durations and for eight different conditions in parallel. To date, this work offers the most comprehensive quantitative view on the interplay between production demand, secretion stress, and effective secretion levels. Moreover, the regulation strategy we propose here is generic since it applies to any protein to secrete and is complementary with classical chassis-engineering strategies.

## Results

### A systematic experimental strategy for characterizing heterologous protein secretion

Our goal is to study the relations between production demand (i.e., induction levels), effective protein production, protein secretion and degradation, and cellular growth for a range of proteins having different secretion complexities. Therefore, we constructed a small collection of yeast strains secreting various heterologous proteins. These proteins differ by their posttranslational modifications (PTMs), sizes and native organisms (Supplementary Note 1). The approach integrates two main components. The first component relies on a dedicated multi-bioreactor platform with automated cytometry measurements and reactive optogenetic control of yeast in continuous cultures[22] (Fig. 1a). The platform is composed of 8 bioreactors operated in parallel. Each reactor is equipped with a set of LEDs to control gene expression and with an optical density (OD) measurement device. To report on the protein production demands effectively perceived by cells subjected to light stimulations, we use an accessory reporter strain, co-cultured with our strain of interest (Supplementary Note 2). In our experiments, media is renewed to maintain the OD at a target level (turbidostat mode). In presence of high secretory burden, the cell growth rate of the strain of interest might drop. Then, the presence of the accessory stain, having a constant growth rate, guarantees a minimal media influx in the reactor. They allow us to infer a good estimate of the growth rate of the strain of interest (Supplementary Note 2). Cells in each bioreactor are sampled and measured by cytometry every 45 min for 24 h thanks to a pipetting robot that connects the output line from the reactors to a tabletop flow cytometer. The second component of the approach consists in a set of genetic constructs that enable a precise control of the production demand, the measurement of

internal levels of the POI together with its secretion-associated stress, and the measurement of secretion levels in the media. The POI is a fusion protein composed of the pre-pro-α-factor secretion signal from *S. cerevisiae*[23,24], the protein under study, the mNeonGreen fluorescent reporter[25], and 3 copies of the FLAG purification tag[26] (Fig. 1b). The fusion of mNeonGreen with the FLAG tags is called the bright tag. Using microscopy imaging, we verified that our bright tag functions as expected, that is, that only the secretion compartments show significant fluorescence in the cell (Supplementary Note 3). To control the production of the POI, we use the EL222 optogenetic gene expression system. This light-oxygen-voltage protein is activated by exposure of the cells to blue light[27]. To inform on the stress produced by the secretion of the POI, we took advantage of previous designs using the UPR as a secretory stress reporter[28,29] by coupling its activation to the expression of a red fluorescent reporter, mScarlet-I[30]. The engineering of our collection of strains is represented in Fig. 1c. In addition, we developed an approach to quantify the secretion levels of the POI in media samples using immunobeads and cytometry (Fig. 1d). We use magnetic beads coated with anti-FLAG antibodies to capture the secreted POI (and only the secreted POI). We show that after proper incubation and several washing steps of the magnetically-retained beads, the fluorescence of the beads is proportional to the quantity of secreted proteins, even at low secretion levels (Supplementary Note 4). This experimental setup constitutes a complete and innovative framework to perform the systematic characterization of the secretion process for various POIs (Fig. 1e). If the external demand is such that the production of the heterologous protein exceeds the secretory capacity of the cell, proteins might accumulate, thereby triggering stress adaptation responses that lead to an increase of the secretory capacity and/or to an increase in protein degradation capacity. Our experimental setup allows us to conduct parallel experiments over long time scales to obtain dynamic data for different secretion-related processes at single-cell resolution, and for different levels of production demand (Fig. 1f).

### Secretion levels may saturate and even decrease with increasing production demand

We characterized the impact of heterologous protein secretion on cell physiology and secretory capacity at different production demands and for different hard-to-secrete proteins. To achieve different levels of production demand using the EL222 optogenetic expression system, we varied the duration of the light exposure within a constant time period of 30 min. That is, we adopt here the duty-cycle encoding of light stimulations as documented in Benzinger and Khammash[31]. Because the light intensities provided by the LED strips might vary from one reactor to another, the effective protein production demand (i.e., induction level) is evaluated using the co-cultured accessory strain that reports the actual production demand perceived by cells. Indeed, these cells express a non-secreted fluorescent reporter under the control of the EL222 transcription factor and integrated into same locus as the POI in the strain of interest.

We monitored the internal POI (iPOI) and UPR levels for our six POIs every 45 min during 24 h in continuous culture and under different production demands. Secretion levels were measured at 24 h. In Fig. 2, we represent the median fluorescence of iPOI and UPR distributions as a function of time. The strain secreting mNeonGreen-3xFLAG (mNeon, Fig. 2, first row) is used to characterize the secretion-associated burden of the bright tag alone. For this strain, we observed that iPOI reaches relatively high levels and then plateaus, whereas the UPR levels remain low. At steady state, iPOI levels and UPR levels increase quasi-linearly with induction strength. Secretion levels are relatively high too. Interestingly, they do not increase linearly with induction strength and iPOI levels, suggesting that, even for a simple fluorescent protein, secretion (and translocation) efficiency is

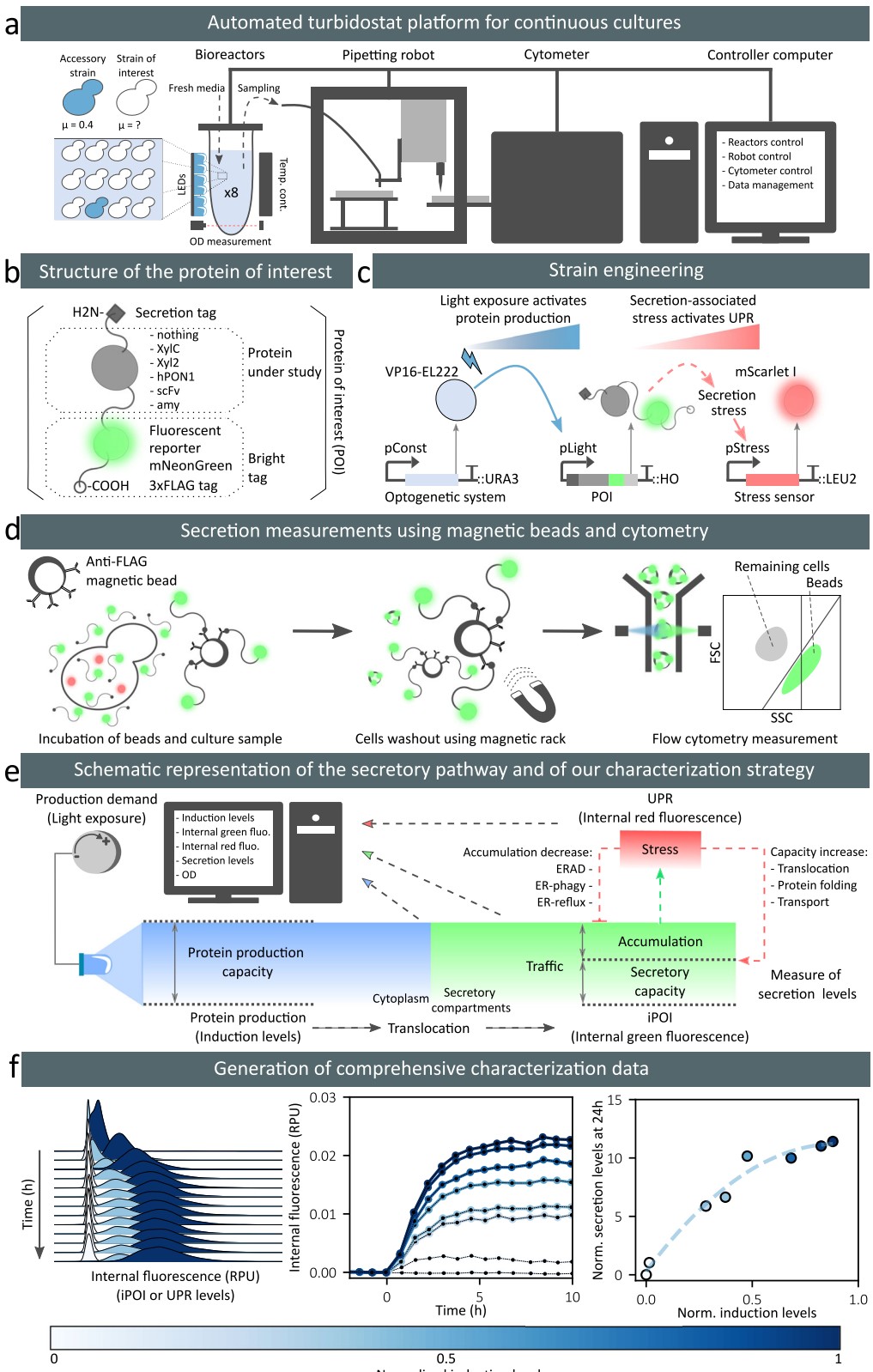

decreasing at high induction levels. In summary and as expected, mNeon appears to be easy to produce and secrete and is not imposing a significant stress on the cell.

Regarding the Endo-1,4-beta-xylanase C fused to the bright tag (XylC, Fig. 2, second row), the iPOI levels and the secreted POI levels were similar to those observed for mNeon. In contrast, the UPR steady state levels were more than twice as high. This shows that, in comparison to mNeon, XylC places a higher load on the secretory pathway but that the cell can adapt its trafficking capacities to compensate for this higher load. This is consistent with the fact that XylC requires a disulfide bond for proper folding to pass the pathway quality control (Supplementary Note 1) and that this PTM is catalyzed by the protein disulfide isomerase (Pdi1), whose expression is modulated by the UPR[12,32,33].

**Fig. 1 | A comprehensive experimental setup for systematic characterization of heterologous protein secretion. a** The platform uses 8 bioreactors for continuous culture of yeast cells. An accessory strain (blue cells) is co-cultured with the strain of interest (white cells) in an initial 1:10 ratio. Output flow lines of reactors are connected to a pipetting robot that prepares the samples and loads them into the cytometer every 45 min. **b** To monitor internal and secreted levels of the heterologous protein under study, the protein is fused to a "bright tag" that comprises of a green fluorescent reporter, mNeonGreen, followed by three copies of the FLAG tag. **c** All modules are integrated in yeast chromosomes. The optogenetic transcription factor EL222 is constitutively expressed (TDH3 promoter, pConst) and activates the pLight promoter upon blue light stimulation. Then, the POI is expressed and may produce secretory-associated stress, activating the unfolded protein response that regulates the expression of genes under the control of the pStress promoter. We introduced a red fluorescent reporter, mScarlet-I, under the control of pStress to monitor secretory-associated stress levels (pUPR). **d** To quantify secretion levels using a cytometer, we developed a methodology based on immuno-magnetic beads capturing only the secreted POI. The distinction between beads and

remaining cells is made thanks to their different light scatter properties (Supplementary Note 4) **e** Schematic representation of our strategy to study protein trafficking and secretion capabilities of cells. The production demand is controlled by blue light illumination of the cell culture. The production of the protein is followed by its translocation into the secretory compartments. Internal protein levels in these compartments (iPOI levels) can be followed by measuring the green fluorescence of cells. The protein can be secreted in the media or can accumulate in the cell. Protein accumulation triggers a secretory stress (UPR stress levels) that can be quantified by measuring the red fluorescence of the cells. The cell stress responses have antagonistic effects with respect to bioproduction: increasing trafficking capabilities or targeting the accumulated proteins for degradation. **f** Example of data obtained by the proposed pipeline. The intensity of the blue color corresponds to the intensity of the light stimulation received by cells in the bioreactor, as reported by the induction level of the co-cultured accessory strain. This corresponds to the protein production demand. RPU: relative promoter units. Source data are provided as a Source Data file.

For the Endo-1,4-beta-xylanase 2 fused to the bright tag (Xyl2, Fig. 2, third row), we observed that for strong inductions the iPOI levels at steady state are half those observed for mNeon and XylC, and that secreted POI levels are comparatively even lower. Moreover, UPR levels are more than twice as high. This indicates that Xyl2 imposes a significant secretory stress and is poorly translocated in the ER, slowly secreted and/or actively degraded. Relating this behavior with the known PTMs is not obvious here. Indeed, since Xyl2 possesses two N-glycosylations, one might have expected that Xyl2 has a longer maturation time than XylC and therefore would accumulate at higher levels.

In the case of the human Paraoxonase 1 fused to the bright tag (hPON1, Fig. 2, fourth row), the data show that for strong inductions iPOI levels at steady state are even higher than the levels observed for mNeon, and that secreted levels are ten-fold lower. Given that hPON1 contains a disulfide bond and three N-glycosylations, these observations are consistent with a slow processing of the protein in the secretory compartments, in connection with its complex PTM needs.

In contrast to all previously studied proteins, the single chain antibody variable fragment 4M5.3 (scFv, Fig. 2, fifth row) and the α-amylase (amy, Fig. 2, sixth row), fused to bright tags, showed pronounced non-monotonic dynamics for their iPOI levels as well as for their UPR levels at high induction levels. Peaks in iPOI and stress levels are observed a few hours after induction. Moreover, scFv and α-amylase-secreting strains showed the highest stress levels (10 to 15 times higher than mNeon at steady state). The burden generated by the secretion of these POIs is also evident when looking at the growth rate dynamics. After 3 h of induction the growth rate decreases by up to 40% of its pre-induction level and then recovers (Supplementary Note 6). Importantly, for the strain secreting scFv, the relation between induction and secretion levels is non-monotonic. This shows the presence of an optimal level of external demand that maximizes protein secretion. From the perspective of protein bioproduction, this secretion sweet spot is important since increasing further the production demand is counter-productive for protein production. The situation appears to be slightly better for the strain secreting the α-amylase (Fig. 2, sixth row). We observed a saturation of the secretion levels for induction above 50% of the maximum but no marked decrease. Yet stress levels continue to increase, showing that high induction levels cause unnecessary stress to the cells.

In summary, this dataset shows markedly different secretion behaviors for the different POIs, confirming that secretory constraints and bottlenecks are protein specific. In particular, we observed induction sweet spots above which cellular secretory stress increases and protein production plateaus or even decreases. Such situations are associated with surprising non-monotonic dynamics in iPOI and stress levels.

## Single-cell data reveal a state of secretion burn-out in a fraction of the cell population

A closer analysis of cytometry data reveals that tracking median levels only can be misleading, since no single cell follows such behavior. Indeed, the non-monotonic dynamics observed for internal protein levels for scFv- and amylase-secreting cells corresponds to the median of a bimodal population. A subset of cells in the population transiently accumulated iPOI at levels that are more than 10 times higher than the other cells (Fig. 3a and Supplementary Note 5). From now on, these cells will be referred to as "accumulators". Formally, iPOI-accumulators, or more simply accumulators, are defined as having iPOI levels higher than the mean plus three standard deviations of a normal distribution fitted to the cell population at 24 h after induction when the iPOI distribution is monomodal again (steady state distribution). When studying the abundance of accumulator cells, defined as the highest fraction of accumulators over the course of the experiment, we saw that accumulators can represent up to 50% of the total cell population and that their abundance increases proportionally to the induction levels above a given threshold. We also observed that the presence of the accumulator cells is correlated with the decrease of the overall growth rate, indicating that these cells have a reduced growth (Supplementary Note 6). This observation is consistent with previous studies describing that in ER-stressed cells, a minimum level of ER functionality is required to complete cell division[34,35]. We note that, in turn, cell cycle issues may exacerbate the accumulation of internal protein in the cells, due to lack of dilution. When cells recover by the action of different adaptive responses, ER capacity and growth rate are restored[34,35]. Lastly, the level of iPOI in accumulator cells is linearly increasing with the induction strength, above a given threshold (Fig. 3a, bottom). This strongly indicates that the secretory pathway and the degradation pathway are both (transiently) saturated in accumulator cells. In summary, when proteins are produced at high levels and cells do not adapt rapidly enough, proteins accumulate in the secretory pathway, overwhelming cell adaptation capacities and impacting growth. These cells are experiencing a secretion burnout. Note that the distribution of UPR levels also appears to be bimodal. The appearance of the UPR bimodal distributions is slightly shifted in time with respect to the appearance of the iPOI one. We will later use the notion of UPR-accumulators to refer to the subset of the cell population that has high UPR levels, using the same definition as for iPOI-accumulators, but based on UPR levels.

The appearance of accumulator cells is a sign that maximal trafficking capacities have been reached. To test the genericity of this observation, we grew cells in different contexts. Firstly, we constrained nitrogen availability in the culture medium, thereby limiting cellular capacities to produce proteins[36,37]. For scFv-secreting cells, we observed that nitrogen limitations strongly reduce the proportion of

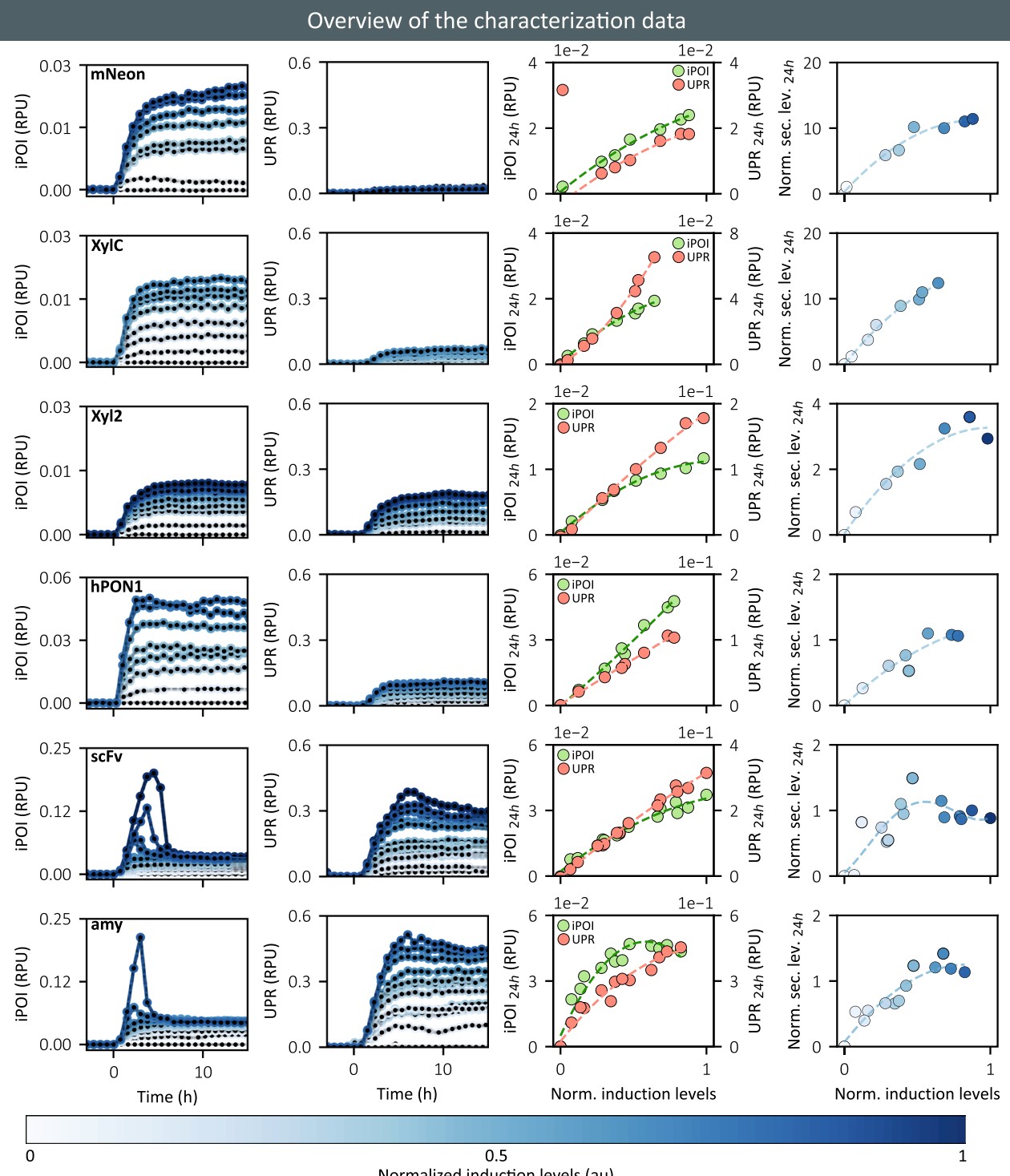

**Fig. 2 | Analysis of protein trafficking and secretion capabilities for a small collection of POIs.** Six strains secreting different POIs were subjected to different production demands. Their iPOI and stress levels were followed in time and secreted protein amounts were quantified at 24 h. Rows represent data for the different proteins (indicated in the top left corner of the first column). Columns represent, for different induction strengths, the temporal evolution of the median levels of iPOI distributions, the temporal evolution of the median levels of UPR distributions, the median iPOI (green) and UPR (red) levels at 24 h, and the secretion levels at 24 h, respectively. Induction levels are color-coded as represented in the bar at the bottom of the figure. iPOI and stress levels are represented here only for the first 15 h. Complete time-courses are shown in Supplementary Note 5. Source data are provided as a Source Data file.

accumulators in the cell population (Fig. 3b, top). Moreover, thanks to the accessory strain, which is also growing in nitrogen-limited conditions, we can estimate the effective protein production rate in these conditions. We found that in normal media or in nitrogen-depleted media, the same protein production rate leads to the same fraction of

accumulator cells. Secondly, we added tunicamycin to the media, a drug that inhibits the N-glycosylations catalyzed in the ER, thereby decreasing the trafficking capacities of the secretory pathway in cells. For Xyl2-secreting cells, we observed that in presence of tunicamycin, accumulators are present even in absence of induction and that their

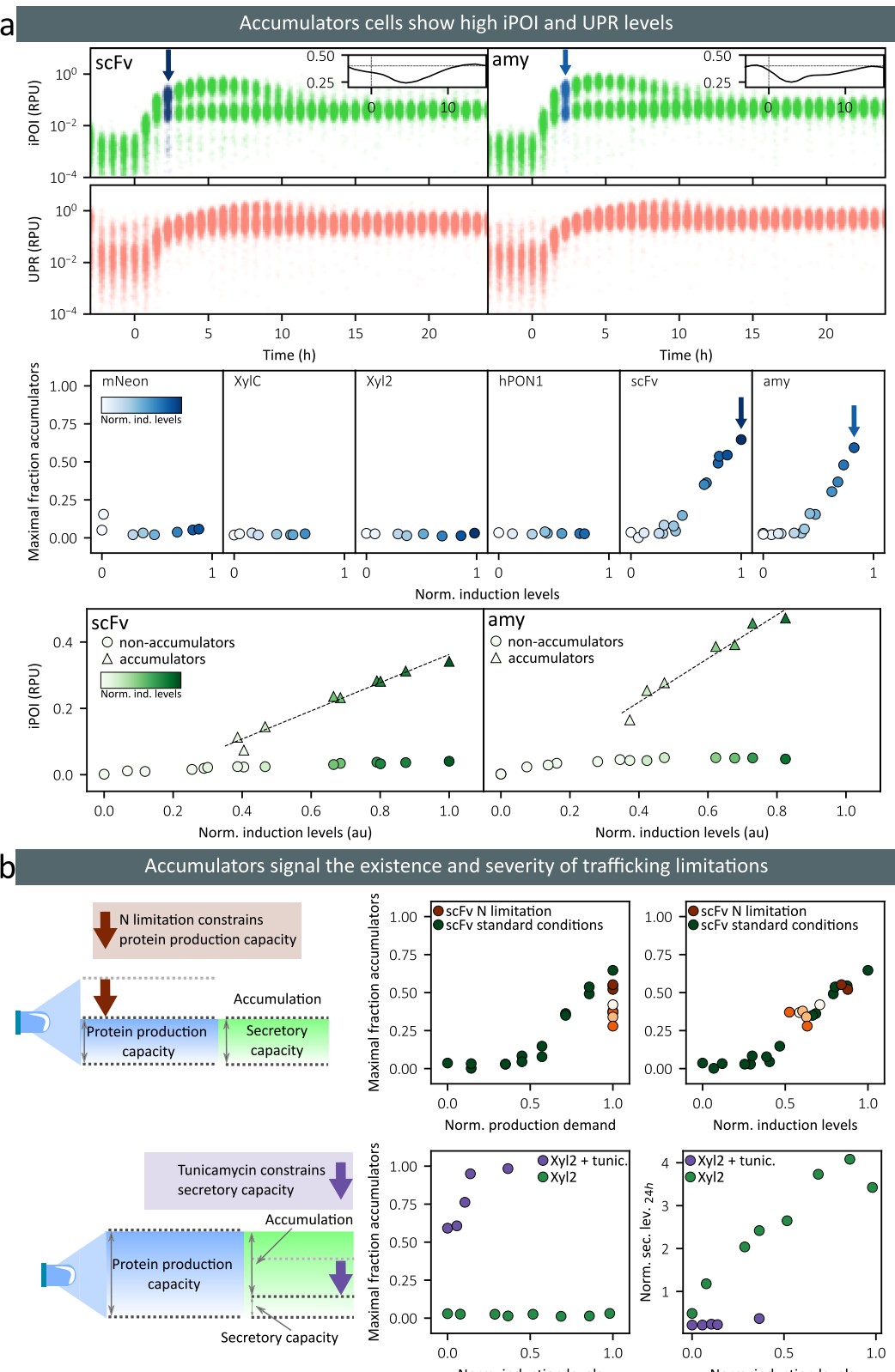

**a** Accumulators cells show high iPOI and UPR levels

**b** Accumulators signal the existence and severity of trafficking limitations

number rapidly increases with induction levels (Fig. 3b, bottom), whereas no accumulators were observed in normal conditions. In summary, we observed for different proteins (ScFv, Amylase, and Xyl2) and in different conditions (normal media, low nitrogen, and presence of tunicamycin) that the presence and the relative fraction of accumulators signals the existence and the severity of trafficking issues in the secretory pathway.

**UPR-mediated adaptation is essential to prevent and recover from secretion burnout**

When confronted with a too strong production demand, cells accumulate proteins to very high levels, stop growing, and experience high stress levels. Yet, this accumulator phenotype is transient and after approximately 5 h, protein levels and stress levels decrease and with time reach the levels of the population of non-accumulator cells. Here,

**Fig. 3 | Accumulators signal that secretory capacities are overflowed. a** At the top, we show single-cell data for iPOI (first row, green dots) and UPR (second row, red dots) at maximal induction levels for scFv (left) and amylase (right). Blue arrows indicate when the fraction of accumulators is maximal. Insets represent the temporal evolution of cell growth rate for the first 15 h of the experiment. The horizontal dotted line indicates the pre-induction growth rate ($0.4\,h^{-1}$). In the middle, we show the maximal fraction of accumulator cells observed over the course of experiments as a function of induction levels. Maxima always appear between 2 and 4 h after induction. The intensity of the blue color in plots is proportional to the induction levels. At the bottom, we show the iPOI levels in the non-accumulator (circles) and accumulator (triangles) subpopulations as a function of induction levels for the secreted scFv (left) and amylase (right) proteins. The intensity of the green color in plots is proportional to the induction levels. **b** At the top, we show the data obtained for the scFv-secreting strain in an experiment in which the

protein production capacity was constrained by limiting nitrogen supply. The maximal fractions of accumulator cells quantified in nitrogen-limited conditions and in normal conditions are shown in orange and green, respectively. The intensity of the orange color corresponds to the ammonium sulfate concentration supplied in each condition (0, 5, 50, 500 or 5000 mg/L). Left and right plots represent the maximal fraction of accumulator cells as a function of the production demand, and as a function of the induction levels, respectively. At the bottom, we show the data obtained in the experiment in which the secretory capacity was limited by adding tunicamycin in the media of the Xyl2-secreting strain. The maximal fractions of accumulator cells in tunicamycin and in normal conditions are shown in purple and green, respectively. Left and right plots represent the maximal fractions of accumulator cells and the secretion levels measured at 24 h as a function of induction levels, respectively. Source data are provided as a Source Data file.

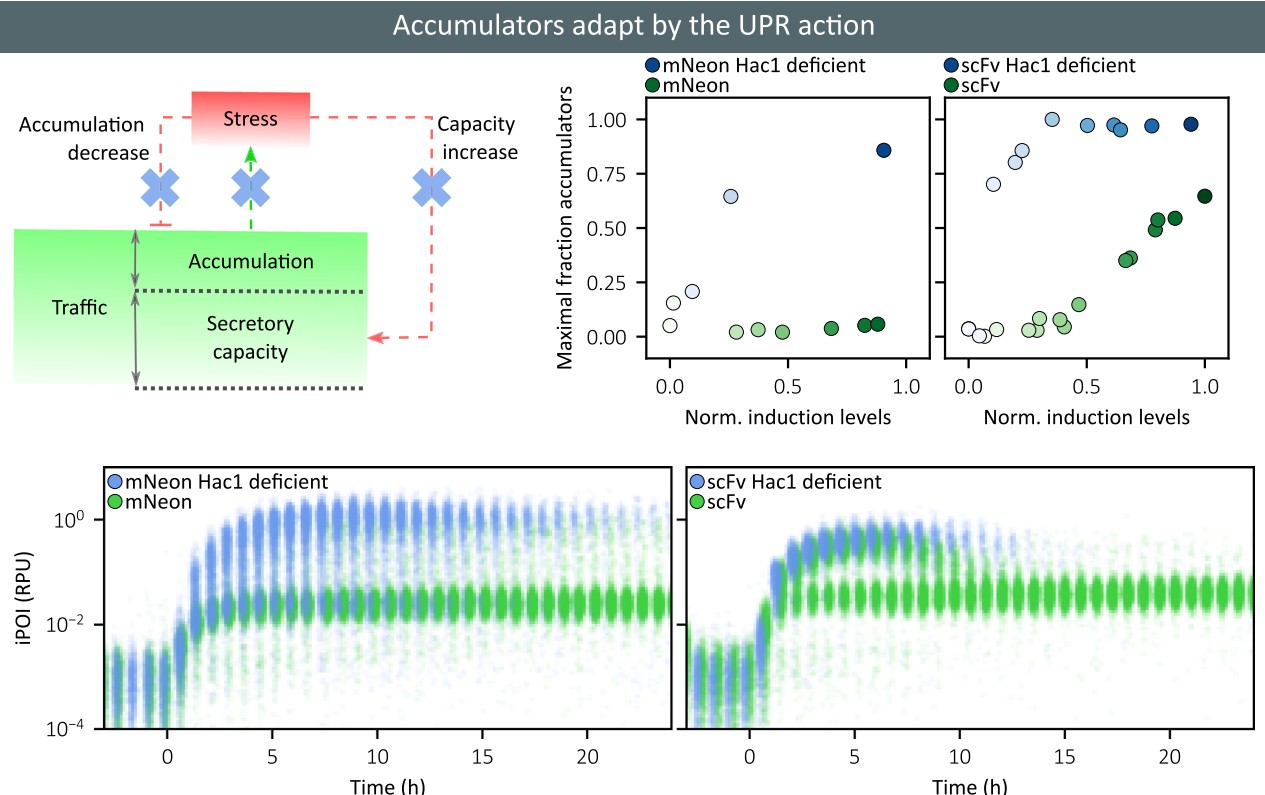

**Fig. 4 | UPR-mediated response is essential for the adaptation of accumulator cells and for maintaining physiology at all induction levels.** At the top, we show the maximal fractions of accumulator cells for Hac1 deficient cells (blue) and for normal cells (green), as a function of the induction levels (left: mNeon-secreting cells, right: scFv-secreting cells). The intensity of the color is proportional to the

induction levels (see scale bars in Fig. 3). At the bottom, we show single-cell data for four representative characterization experiments with maximal induction levels (blue dots: knockout strain, green dots: wild-type strain; left: mNeon-secreting cells, right: scFv-secreting cells). Source data are provided as a Source Data file.

we investigate the role of the stress adaptation pathway in this rather spectacular adaptation. To do so, we constructed Hac1 deficient strains. Hac1 is the transcription factor that binds to the promoter controlling the expression of UPR-regulated genes[12,38]. Therefore, Hac1-deficient cells are not capable of triggering the UPR.

The *HAC1* knock-out mutant of the scFv-secreting strain showed a very severe phenotype. The entire cell population shows the accumulator phenotype, stops growing, does not recover and gets eventually diluted away from the bioreactor (Fig. 4, Supplementary Note 7). This observation demonstrates that UPR-mediated adaptation is the principal, if not unique, mechanism of adaptation for cells experiencing secretion burnout. More surprisingly, accumulators also appeared for the *HAC1* mutant of the mNeon-secreting strain. Their fraction is low at low induction levels but rapidly increases with increasing induction levels. This reveals that in *S. cerevisiae*, the

nominal trafficking capabilities are exactly matching the needs of the native secretome and that imposing additional demands will trigger adaptation responses.

## The appearance of accumulators signals the activation of the degradation pathway in the entire cell population

We have previously identified that some hard-to-secrete proteins, namely scFv and amylase, present induction sweet spots (Fig. 2). The efficient production of the proteins necessitates that the induction of gene expression is made at carefully chosen levels. These levels themselves depend on the cell environment. Scanning induction levels and quantifying secreted protein levels in a systematic manner is highly time consuming. In Fig. 5a, we show that secretion sweet spots can instead be identified by tracking the appearance of accumulators in the cell population. Indeed, the induction levels that maximize

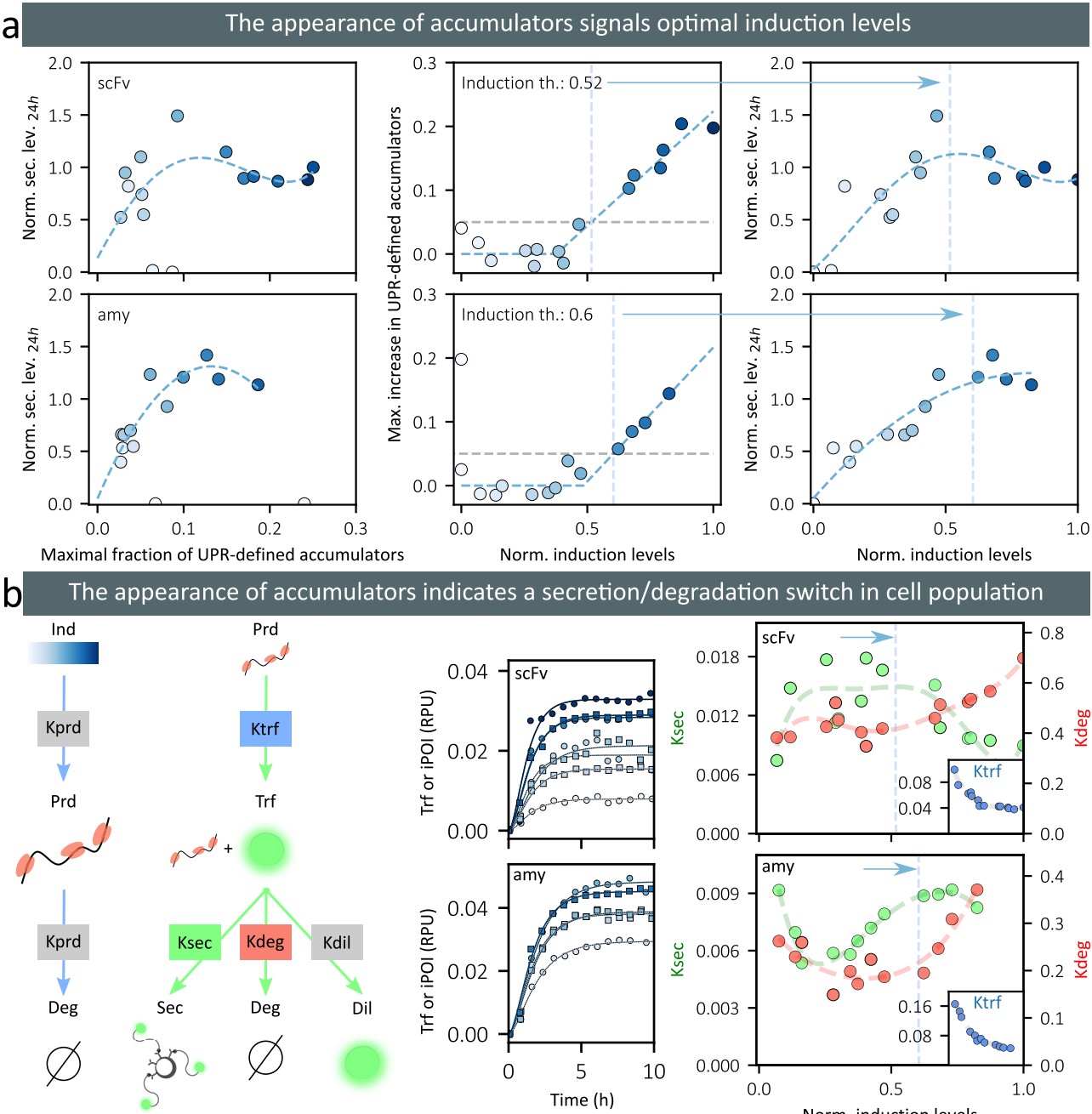

**Fig. 5 | The appearance of accumulator cells indicates secretion sweet spots where the balance between protein secretion and degradation is optimal. a** On the left, the plots show the relationship between the occurrence of (UPR-)accumulator cells and the protein secretion levels at 24 h (top: scFv-secreting strain, bottom: amylase-secreting strain). In the middle, the plots represent the maximal fraction of UPR-accumulators as a function of the induction levels. The induction thresholds indicate the levels of induction at which the net increase of UPR-accumulator cells reaches 5% of the population (blue dotted bars). On the right, the plots represent the secretion levels at 24 h as a function of the induction levels (same as in Fig. 2), with an indication of the previously defined induction thresholds (blue dotted bars). We observe that they correspond to maximal secretion levels. The intensity of the blue color in dots corresponds to the induction levels. **b** On the left, we represent our simple mathematical model focusing on the production and translocation in the trafficking compartments of the protein, and its possible outcomes: secretion, active degradation, or dilution due to cell growth. Parameters in gray boxes are calibrated using independent experiments and have fixed values. The other parameters are calibrated using time series and secretion data, independently for each induction level. In the middle, the plots represent the temporal evolution of the iPOI levels in the non-accumulator cell population together with model fits (top: scFv-secreting strain, bottom: amylase-secreting strain). On the right, the plots represent fitted parameter values as a function of induction levels (Ksec: green, Kdeg: red, and Ktrf: blue in inset; top: scFv-secreting strain, bottom: amylase-secreting strain). Blue dotted bars indicate the previously defined induction thresholds. Source data are provided as a Source Data file.

protein production correspond to the appearance of accumulators, where appearance is defined as an increase of 5% above basal level of the maximal fraction of accumulators. Note that we define accumulators here with respect to UPR levels instead of iPOI levels (bimodal distributions are observed in both cases). This definition is more convenient since this criterion is protein independent and does not require complex genetic constructions. It can be tested in various application contexts. The maximal fractions of iPOI-defined accumulators and of UPR-defined accumulators are linearly related (Supplementary Note 5).

We developed a simple mathematical model to better understand the cell physiology around these production sweet spots. We focused on quantifying the relative impact of the adaptative responses that increase trafficking capacities or that target proteins for degradation. Distinguishing these antagonistic effects is essential for understanding production efficiency. Because the accumulator population is marginal around secretion sweet spots and accumulators are present only transiently, we focused on understanding the adaptation response of the non-accumulator cells. The model is based on ordinary differential equations and captures three main processes with simple assumptions. A protein production intermediate, Prd, is produced proportionally to the optogenetic induction level, and is degraded at a rate proportional to its concentration. This intermediate can typically be the messenger RNA of the POI. It is important to account for the initial delay of protein appearance. This Prd intermediate is used to produce a protein that is translocated in the secretory compartments. This "protein in traffic", Trf, is then either secreted, actively degraded (by ERAD for example), or diluted in the cell because of cellular growth. The model also accounts for the secretion of the proteins by cells in the media, and their dilution due to media renewal at a rate that equals the cell growth rate (turbidostat mode). To deal with parameter non-identifiability at the transcriptional level, two parameters are set equal (Kprd). Moreover, to simplify parameter identification, the model is first fitted to experimental data obtained for a strain expressing a non-secreted mNeonGreen protein (Ksec = Kdeg = 0). Parameters values obtained for Kprd and Kdil were consistent with our experimental conditions and with literature values[39,40] (cell generation time of 90 min and mRNA half-life of 20 min). The model was then independently fitted to each time-series to identify how the parameter values evolve as a function of the production demand. The model, together with the data for the non-accumulator cell population, model fits, and the corresponding parameter values are represented in Fig. 5b. For both POIs, the parameter estimates show that protein degradation rates increase when induction exceeds the accumulator-appearance thresholds defined previously. At high induction levels, protein secretion rates either plateau (amylase) or decrease (scFv). In summary, our quantitative analysis reveals that at the induction levels where accumulators appear, cells have reached their maximal secretory adaptation capabilities. Increasing the demand further leads to increasing protein degradation in the non-accumulator cells or transiently experiencing a secretion burnout in accumulator cells.

**Maximizing protein production using real-time control and optimal stress levels**

The results of the previous section can be used to propose a real-time control strategy to maximize protein bioproduction. Firstly, one identifies what should be the optimal level of stress, that is the UPR stress level that is associated with the appearance of accumulators. This should indicate a physiological state in which cell adaptation is optimal for secretion. As before, we define this target stress level as the stress felt by cells when light induction was such that the maximal fraction of accumulators is 5% above baseline levels (Fig. 6a). We use mean stress levels instead of median values so that our control strategy can be implemented using simpler measurement devices than cytometers, such as plate-readers for example. We then propose a very simple control strategy. When UPR levels were below the reference level, the duration of light stimulation is increased by a fixed amount (5% of sampling period, that is, 2 min and 15 s given that the sampling time is 45 min), and conversely, when UPR levels were above the reference, the duration of the light stimulation is reduced by the same amount (Supplementary Note 8). We start with no light. Here, we take advantage of the real-time control capabilities of our experimental platform, which has been previously employed to achieve real time control in diverse experimental contexts[22,41,42]. Several real-time control experiments were performed using reference UPR stress levels in

the vicinity of the previously defined target level, and protein production by scFv-secreting cells was quantified in the media. We found that secretion levels were indeed maximal when the effective UPR levels were maintained close to the target level (Fig. 6b). Moreover, we found that the maximal level is 70% higher than the level of secreted protein one obtains using a constant, full-light induction (Fig. 6b). This experiment demonstrates that by tracking the appearance of accumulator cells, one can define a target stress level that leads to optimal protein production.

## Discussion

We demonstrated that non-trivial optimal demand levels exist for hard-to-secrete proteins. These induction sweet spots are protein- and condition-dependent. Increasing the demand on protein production beyond these levels leads to imposing more stress on the cells with no more or even less secreted proteins. It is also accompanied by the surprising appearance of a transient bimodal response of the cell population in which a subset of the cells accumulates high amounts of proteins, decreases growth, and faces significant stress, that is, experiences a secretion burnout. The signature phenotype of the accumulator cells is observed when the production demand reaches the maximal trafficking capabilities of the cell and the adaptation response is too weak or too slow. The maximal fraction of accumulator cells observed in the population quantifies the impact of the secretion burden at the cell population level. We also show that in secretion burnout situations, cell adaptation is largely, if not exclusively, mediated via the UPR response. Using a quantitative analysis, we found that when accumulator cells appear, protein degradation rates start to increase in the main cell population (the non-accumulator cells). The presence of accumulator cells is therefore a single-cell phenomenon that reflects a population effect. Given that the accumulator phenotype is visible not only in internal POI levels but also in UPR stress levels, we tested a regulation strategy in which first we identified the induction level at which UPR-accumulators appear, and second, we use real-time control to maintain the cell population in the vicinity of this stress level. Using this strategy, we obtained up to 70% higher levels of secreted proteins than what is obtained using a maximal induction strategy. We focused on scFv-secreting cells for our control experiments since the highest secretion level is obtained for these cells at intermediate induction levels. Because this feature is not shared by the other proteins we tested in this work, we do not expect to obtain significant benefits for the optimization of the secretion of the other proteins. We also note that it is possible that a carefully chosen open loop control strategy can perform equally well than the closed loop control experiment we show. Setting up this open loop strategy amounts to finding via trails-and-errors the induction level that triggers a stress that is close to the target stress level used in our closed loop control experiment. The use of a closed loop control strategy is an alternative to this trial-and-error approach for open loop control. Close loop control strategies might therefore be more time efficient. Moreover, the relation between induction and secretion stress is protein-dependent and media-dependent. We anticipate, although we do not demonstrate it, that closed loop control strategies offer more versatile and robust solutions than those based on open loop control. Lastly, we stress that the regulation strategy we propose here, based on real-time measurements of secretion stress levels in cells, applies in principle to any protein to secrete and is complementary with classical chassis-engineering strategies. It only requires the addition to the cells of a secretion stress level reporter. No modification of the secreted protein is needed.

The combined use of two components has been instrumental in obtaining these results. The first one is the design and construction of a small collection of strains allowing to conveniently quantify the internal levels of the protein of interest, the secretion stress levels, the cellular growth rates, and the protein secretion levels in media, for a

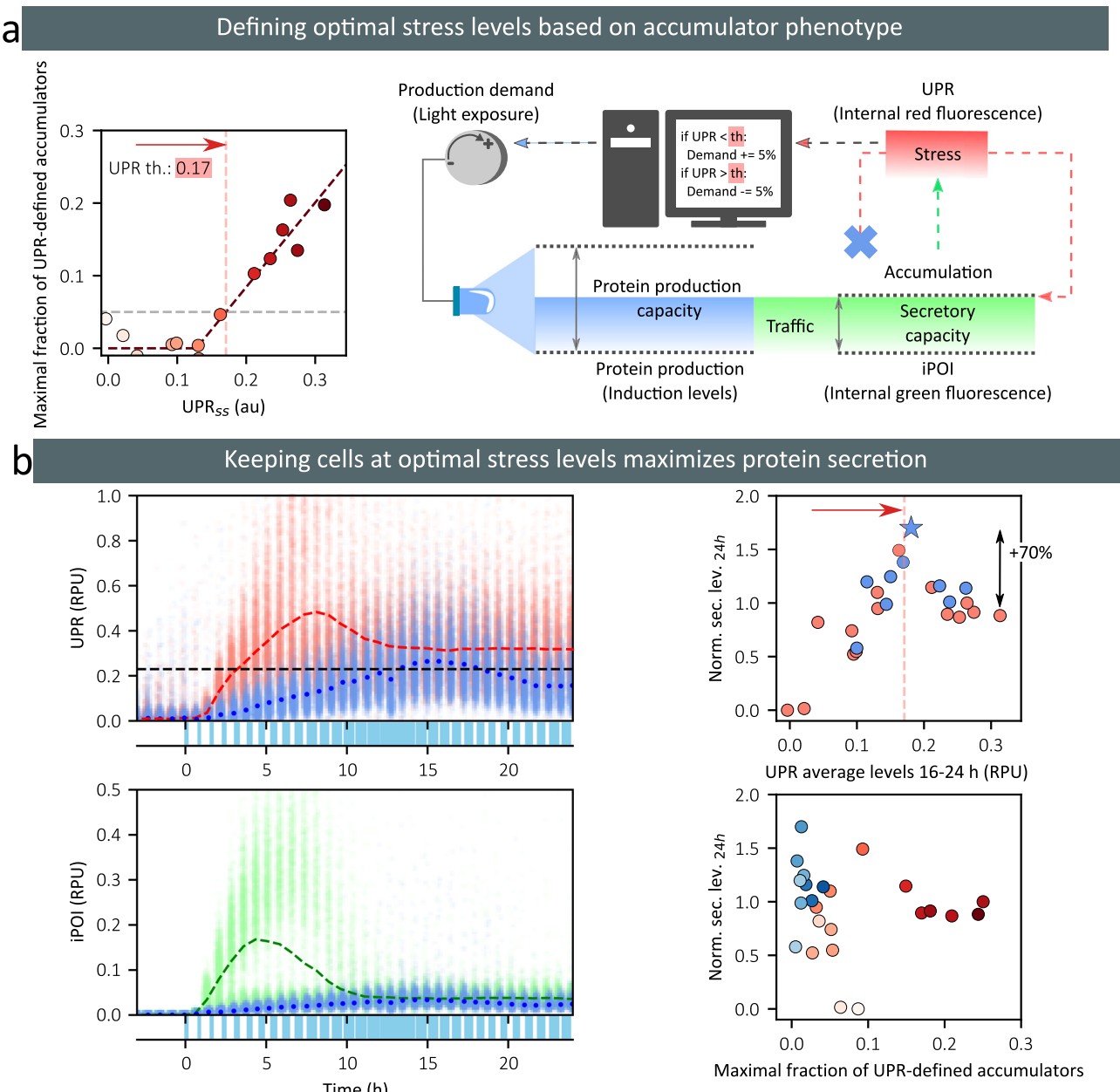

**Fig. 6 | Real-time control approaches that keep cells at optimal stress levels maximize secreted protein levels. a** On the left, the plot shows the maximal fraction of UPR-defined accumulator cells as a function of the mean stress level of the population at 24 h, for scFv-secreting cells. The target stress level is defined as the UPR level at which the net increase of UPR-accumulator cells reaches 5% of the population. On the right, our control strategy that aims at maximizing productive adaptation and minimizing deleterious adaptation to secretory stress is represented. The average levels of UPR are monitored every 45 min, and depending on their values with respect to the reference value (th), the production demand is increased or decreased by 5% of the stimulation period (45 min). **b** On the left, the single-cell data for UPR (top) or iPOI (bottom) levels in scFv-secreting cells are represented as a function of time in a full-induction experiment (red and green dots) or in a real-time control experiment (th = 0.23, light blue dots). Mean values are represented by dashed (red and green) lines for the full-induction experiment, and by dark blue dots for the real-time control experiment. Light stimuli for the real-time control experiment are represented at the bottom of plots (blue rectangles). In this experiment, the average of the UPR levels during the last 8 h of experiment equals 0.17. On the right, we represent the secretion levels obtained in different real-time control experiments (blue dots) and in different characterization experiments (red dots) as a function of the average UPR levels over the last 8 h of the experiment (top) or as a function of the maximal fraction of UPR-defined accumulators (bottom). The blue star corresponds to the secretion levels of the real-time control experiment shown on the left. In the top plot, the target level (i.e., the optimal stress level defined in the panel above) is shown by a red bar. In the bottom plot, the intensity of the color corresponds to the average UPR levels over the last 8 h of experiment. Source data are provided as a Source Data file.

range of proteins of different secretion complexities and for a range of induction levels. The second is the use of optogenetically-enabled bioreactor setup with automated sampling and cytometry measurements. This setup allowed to obtain single cell measurements and growth rate measurements at a high temporal resolution for 8 cultures in parallel and for extended durations. We also took advantage of

magnetic immunobeads assays for the systematic measurement of secretion levels for a set of proteins.

Naturally, we are not the first ones to report the existence of induction sweet spots. This is a long-known fact[43,44]. Yet, by combining synthetic biology and lab automation approaches, we provide here the most comprehensive quantitative view on protein secretion in yeast.

Moreover, we identified a surprising transient bimodal response of the cells with a specific burnout state, and demonstrated its relevance for the identification of optimal induction levels. Importantly, our proposed regulation strategy is based on simple measurements of mean UPR stress levels in cells and can therefore be applied independently of a specific genetic background. In particular, it is in principle compatible with and complementary to the extensive chassis-engineering optimization strategies found in industrial applications. We note that the presence of bimodal distributions for secretion-associated features in yeast has already been described in previous works using microfluidics platforms[45]. Yet, their significance for optimizing protein production was not discussed. Lastly, this work also demonstrates that cybergenetic approaches can effectively be used to optimize cellular processes of interest. Many proofs of concept of the quantitative regulation capabilities of cybergenetic approaches have been provided to date (e.g., refs. [46–52]). Yet, only a few works have demonstrated that some process of interest can actually be improved thanks to real-time control strategies[53,54]. Interestingly, Benisch and colleagues presented a closely-related work in a recent publication[55]. Specifically, they observe that higher levels of secreted α-amylase can be obtained using optogenetic real-time control approaches that keep cellular UPR stress at intermediate levels. This is not what we observed in this study. In our case, we do not observe significant secretion defects, but we do observe significant transient growth defects for the α-amylase at high induction levels. This discrepancy can be due to differences in experimental setups. Benisch and colleagues use a 1 L bioreactor in batch mode, and a two-phase control strategy in which a real-time control phase is used for 24 h, followed by a full induction phase lasting until 70 h, when amylase activity is measured. It is therefore possible that the highest secretion levels measured at the end of the experiment for intermediate induction levels originate from a more regular growth rather than an improved secretion. Indeed, full-light induction causes a 10-fold decrease in optical density at 24 h in the batch bioreactor[55]. The observed differences can also originate from the fact the strains and genetic constructions are not exactly the same. Moreover, Benisch and colleagues observed like us a non-monotonic behavior of the stress level in cells as a function of time. In their case, this behavior is explained by means of an elaborate model of ordinary differential equations (10 state variables, 26 parameters), whereas we explain it by the transient presence of cells in secretion burnout.

The first obvious direction for future works is to investigate the mechanistic origins of the bimodal response of a population of cells subjected to a strong demand in protein production and secretion. It is likely that the observed differences originate in preexisting differences in cell capacities to activate protein degradation as soon as protein starts to accumulate in the secretion pathway. Yet, the molecular mechanisms remain to be clarified. A second direction for future works is to design and construct genetic circuits that internalize the feedback loop so that each cell tunes its response to the external demand based on its own stress level. Such regulation strategies have already been proposed in different contexts[56–58]. Regarding secretion optimization, the identification of stress responsive promoters that can be appropriately used to close the regulation loop remains a challenge. The recent characterization of novel biosensors for secretory stress will provide a sound starting point[59].

## Methods

### Cloning and strains construction

The genetic constructions used in this study were designed to be compatible with the Yeast Tool Kit modular cloning strategy developed by Lee and colleagues[60]. The MoClo-YTK plasmid kit was a gift from John Dueber (Addgene kit # 1000000061). Some parts corresponding to coding sequences or promoters were purchased from external suppliers, Twist Biosciences or GENEWIZ, depending on the

specific needs. The coding sequences of the proteins of interest were obtained from literature and verified in GeneBank (Supplementary Note 1). These sequences have been codon-optimized for expression in *S. cerevisiae*. In all cases, the final vectors are integrative plasmids for yeast (Supplementary Note 1). Transformations were based on an adapted version of the lithium acetate/single-stranded carrier DNA/PEG method from Gietz and colleagues[61,62].

All strains are derived from the common *S. cerevisiae* laboratory strain BY4741. Strain details are described in Supplementary Note 1. All strains used in this work express the light-inducible transcription factor EL222[27] from the *URA3* locus (transcriptional unit: pTDH3 NLS-VP16-EL222 tSSA1). In all cases, the POI expression cassettes are integrated in the *HO* locus and confer constitutive expression of *HIS3* for positive selection. The transcriptional unit includes the light inducible promoter, composed of five copies of the pC120 EL222-binding sequence and the minimal promoter CYC180[31], followed by the POI coding sequence (transcriptional unit: pC120x5-pCYC180 POI tTDH1). The ER-associated stress reporter is adapted from Pincus and colleagues[28]. Our pStress promoter is a hybrid promoter that combines a crippled *CYC1* promoter with 4 repeats of a UPR-responsive element UPRE[29]. The red fluorescent protein mScarlet-I controlled by pStress is integrated in the *LEU2* locus (transcriptional unit: UPREx4-pCYC180 mScarlet-I tENO1). To generate the *HAC1* knock-out, we used a CRISPR/Cas9-based[63] method to introduce an early 5′ TAG amber STOP codon replacing the protospacer adjacent motif (PAM) within the coding sequence of the target gene. All strains used in this study were sequenced for the proper integration of expression cassettes.

### Culture conditions for characterization experiments

All characterization experiments were performed in 20 mL of culture volume in the bioreactors at 30 °C, in turbidostat mode (OD 0.5, typically corresponding to $10^7$ cells/mL according to cytometry data). Details on the bioreactors and the experimental platform can be found in ref. [22]. The media used was synthetic complete (Formedium LoFlo yeast nitrogen base CYN6510 and Formedium complete supplement mixture DCS0019), 2% glucose (w/v) and 5 mM of L-arginine (Sigma 11009). L-arginine is used to maintain pH at 7, thus keeping secreted protein integrity along the duration of the experiment[64–66]. After an overnight (16 h approx.) within the bioreactors, the automated cytometry measurements started. Samples were taken every 45 min and 5000 events per sample were recorded by cytometry. After 3 to 4 h of measurements in dark conditions, the LEDs of the different reactors were switched on for various durations within periods of 30 min during 24 h. To provide reliable and accurate measurements of the growth rate and induction levels, as well as to guarantee a minimal media flow through the reactors in case of a severe reduction of the growth of the strain under study, a yeast accessory strain was co-cultured with the strain of interest at an initial ratio of 1:10 in all experiments (Supplementary Note 2). All experiments were protected from direct light.

### Culture conditions for nitrogen-limiting experiment

For the nitrogen-limiting experiment the media used was synthetic complete without ammonium sulfate (Formedium LoFlo yeast nitrogen base without ammonium sulfate CYN6210 + Formedium complete supplement mixture DCS0019), 2% glucose (w/v) and 5 mM of L-arginine (Sigma 11009). Then, the ammonium sulfate was added (Sigma ammonium sulfate A4418) to reach final concentrations of 5, 50, 500 and 5000 mg/L. In this experiment, the OD was dynamically maintained between 0.4 and 0.6. The culture was kept growing during an overnight in our standard media. Then, we changed to nitrogen-limiting media and started cytometry measurements. Eight hours after, the optogenetic induction with constant light was triggered for 24 h. Cytometry sampling frequency and settings, LEDs intensity, use of the accessory strain, temperature, and culture volume were similar to the characterization experiments.

## Culture conditions for tunicamycin stress-induced experiment

After an overnight growth in our standard media, we change to a media containing tunicamycin in DMSO (Bio-Techne 3516) at 0.25 mg/ml final concentration and started cytometry measurements. Eight hours after, the optogenetic induction with constant light was triggered for 24 h. All other settings were similar to those used in the characterization experiments. Previous characterization experiments were done to decide on the concentration of tunicamycin and the DMSO effects on the cell culture.

## Culture conditions for control experiments

In the control experiments, all settings and media were similar to those used in the characterization experiments, with the exception that the accessory strain was not used (with one exception) because we wanted to minimize the probability of counting red fluorescence from the sensor strain as UPR signal. After an overnight of growth within the bioreactors, the automated cytometry measurements started sampling every 45 min. After 3 to 4 h of measurements in dark conditions, the LEDs of the platform were switched with duty cycles set by the control strategy. Note that in feedback experiments, the light stimulation period was 45 min to match the sampling period. Three sets of 8 experiments have been produced in different days, each of them including one experiment in which the maximal demand strategy was applied. These experiments were used as reference to normalize secretion levels and maximal fractions of accumulators across the different sets of experiments.

## Accessory strain for assessing effective induction levels and quantifying growth rates

To provide reliable and accurate measurements of the growth rate and induction levels, a yeast accessory strain was co-cultured with the strain of interest at an initial ratio of 1:10 (Supplementary Note 2). To allow differentiating one yeast strain from the other in co-cultures, the accessory strain constitutively expresses a blue fluorescent reporter (transcriptional unit: 2x[pTDH3 mCerulean tTDH1], integrated in the *LEU2* locus and conferring *LEU2* constitutive expression). Moreover, the accessory strain expresses a cytoplasmic red fluorescent protein, whose expression is controlled by the EL222 optogenetic system as for the gene of interest in the strain under study (transcriptional unit: pC120x5-pCYC180 mScarlet-I tTDH1, integrated in the *HO* locus and conferring *HIS3* constitutive expression). Therefore, at steady state, expression levels of the red fluorescent protein in the accessory strain inform on the induction levels of the POI in the strain of interest. Normalized induction levels are defined as the red fluorescence intensity of the co-cultured accessory strain divided by the maximal red fluorescence intensity of the accessory strain found across all experiments shown in this study. Therefore, normalized induction levels can be directly compared across experiments. Moreover, we use the accessory strain to assess the growth rate of the strain of interest. Indeed, the dynamics of the relative amount of the two strains allows us to infer the growth rate difference for each condition (Supplementary Note 2).

## Cytometry data analysis

To account for minor day-to-day variability and settings adjustments for the cytometer (Guava EasyCyte 14HT, Luminex), we computed a correction coefficient for all the channels used in each experiment. Such coefficient was computed by using the *Guava® easyCheck™* Kit, in which fluorescent beads are measured in every channel, providing the changes in mean fluorescence intensities from one experiment to another. These coefficients are then used to normalize the fluorescence obtained from the different channels in each experiment. The channels used for each fluorescent reporter were mNeonGreen: GRN-B channel, mCerulean: BLU-V channel and mScarlet-I: ORG-G channel. Gating is used on the forward light scatter (FSC) channel to discard cell doublets and cell debris (only events having values between 1000 and 2000 in FSC were

kept). After this FSC gating, we differentiated the accessory strain from the strain of interest using their mCerulean fluorescence. We considered events with BLU-V signal above $6 \cdot 10^{-2}$ as cells from the accessory strain cells and events with BLU-V signal below $3 \cdot 10^{-2}$ as cells from the strain of interest (Supplementary Note 2). To convert raw cytometry data into fluorophore concentrations in relative promoter units[67] (RPU), the fluorescence of each event in each channel was divided by its FSC to yield size-normalized fluorophore levels, and by the fluorescence of cells expressing the same fluorophore under the control of pTDH3. Finally, we subtract the mean fluorescence in RPU units during the 2 h before light induction to all data points to avoid accounting for promoter leakage and autofluorescence. All the analysis for data processing is done using *Python 3*, with *pandas*[68], *numpy*[69], and *SciPy*[70] packages.

## Secretion measurements and analysis

For measuring the secretion levels, we used 4% agarose magnetic micro-beads bound to the Anti-FLAG M2 monoclonal antibody (Anti-FLAG® M2 Magnetic Beads from Sigma, M8823). Beads size ranges between 20–75 μm, which allowed us to use them in the cytometer. The protocol uses 0.5 μL of packed gel volume of beads per sample. After the beads equilibration (following the vendor recommended procedure), we collected the beads using the magnetic rack of the pipetting robot and resuspend them in 10 μL of water per assay. Then, 200 μL of culture samples were mixed with 26 μL of phosphate buffer 1 M and 10 μL of the equilibrated beads. In addition, all our bead measurements contained 26 μL of mCerulean-3xFLAG at approximately 1 μg/ml (not used for analysis here). After incubation during 1 h at room temperature, the samples were washed three times before measurements by collecting the beads with the magnetic rack and resuspending them in 200 μL of TBS buffer. All manipulations with samples containing cells were done in the dark to avoid inducing the optogenetic system expressing the POI. Finally, bead solutions were passed to the cytometer and up to 1000 events were collected.

As for the yeast fluorescence measurements, we applied to beads the correction coefficients obtained using the *Guava® easyCheck™* procedure. Despite the washing procedure, some cells were still present in the sample. We applied two gating conditions based on minimal side scatter (SSC), and minimal SSC/FSC scatter ratio. Then, to assess the secretion levels of the population in each reactor, we computed the median fluorescence on the GRN-B channel. To subtract autofluorescence and the residual crosstalk from the mCerulean-3xFLAG traces in the sample into the GRN-B channel, we used the beads corresponding to the non-induced sample as a blank. Thus, we subtract the median florescence of the non-induced sample to all the beads samples in each experiment. Additionally, since we used the accessory strain which abundance is changing in time, we normalized the secretion levels by the fraction of cells of interest in the population, corresponding to the fraction of cells that actually secrete. A label free proteomics analysis was used to obtain scaling factors for bead fluorescence to obtain measurements that are comparable for the different proteins (Supplementary Note 4).

## Mathematical model

We used an ordinary differential equation model having three state variables and six parameters.

$$\frac{d\mathrm{Prd}}{dt} = K_{\mathrm{prd}} \cdot \mathrm{Ind} - K_{\mathrm{prd}} \cdot \mathrm{Prd} \tag{1}$$

$$\frac{d\mathrm{Trf}}{dt} = K_{\mathrm{trf}} \cdot \mathrm{Prd} - \left( K_{\mathrm{sec}} + K_{\mathrm{deg}} + K_{\mathrm{dil}} \right) \cdot \mathrm{Trf} \tag{2}$$

$$\frac{d\mathrm{Sec}}{dt} = K_{\mathrm{sec}} \cdot K_{\mathrm{units}} \cdot \mathrm{Trf} - K_{\mathrm{dil}} \cdot \mathrm{Sec}, \tag{3}$$

where Prd, Trf, and Sec correspond to production intermediates, proteins in traffic, and secreted proteins, respectively. Two parameters (Kprd and Kdil) were fitted to data from cells expressing a non-secreted mNeonGreen protein and then kept constant. The same parameter was used for cell growth and media renewal (Kdil) because the model captures only the behavior of non-accumulator cells that grow normally and because the bioreactors are operated in turbidostat mode. The Kunits parameter is a scaling factor between internal and secreted fluorescence values. It was set based on secreted mNeon-Green data. The remaining three parameters were fitted on individual time series data (Trf levels) and on secretion data (Sec levels) for each induction level and each protein based on the non-accumulator cell population data. The non-accumulator and accumulator cell populations were distinguished using a Gaussian Mixture Model (Supplementary Note 5, Python package *sklearn.mixture.GaussianMixture*[71]). The ODE model was integrated using the numerical solver *scipy.integrate.solve_ivp*[70], and parameter fitting was performed thanks to the CMA-ES algorithm using the *pycma* package from Hansen and colleagues[72].

### Statistics and reproducibility
No statistical method was used to predetermine sample size. No data were excluded from the analyses. The experiments were not randomized. The Investigators were not blinded to allocation during experiments and outcome assessment.

### Reporting summary
Further information on research design is available in the Nature Portfolio Reporting Summary linked to this article.

## Data availability
All the raw experimental data generated in this study have been deposited on Zenodo (https://doi.org/10.5281/zenodo.7418639) and Pride PXD041650. Sequences of plasmids used to construct all yeast strains are available in the GenBank format in the YeastCyberSecretion Inria Git repository (https://gitlab.inria.fr/InBio/Public/yeastcybersecretion). Source data are provided with this paper.

## Code availability
To grow cells, control gene expression, take samples and measure cell fluorescence by cytometry, we used an automated platform driven by the ReacSight software available at https://gitlab.inria.fr/InBio/Public/reacsight. ReacSight operations are described in a Jupyter notebook associated with each experiment. ReacSight notebooks are available online at https://doi.org/10.5281/zenodo.7418639. In addition to ReacSight (v1), we also used the OT-2 Python API (v2), and GuavaSoft (v3.3). The Python code to process and analyze the raw data and generate figures for the manuscript can be found at https://gitlab.inria.fr/InBio/Public/yeastcybersecretion. It uses pandas, numpy, scipy, sklearn, cma and matplotlib packages.

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

## Acknowledgements

The authors would like to thank Chetan Aditya, Achille Fraisse, Allyson Holmes, Hélène Philippe, and Jakob Ruess for helpful discussions on cloning procedures or model development. We thank Mariette Matondo and Thibault Chaze of the proteomics facility of Institut Pasteur for LC-MS/MS protein quantification. This work was supported by ANR grants

CyberCircuits (ANR-18-CE91-0002; S.S.-C.), MEMIP (ANR-16-CE33-0018, S.S.-C.), and SmartSec (ANR-21-CE44-0033, G.B.), by the H2020 Fet-Open COSY-BIO grant (grant agreement no. 766840, S.S.-C.) and by the Inria IPL grant COSY (G.B.).

## Author contributions

S.S.-C., F.B., and G.B. conceived the study. S.S.-C. constructed all strains except the two HAC1 knock-out strains, performed all experiments, analyzed data, and developed mathematical models and control procedures. H.G. constructed the two HAC1 knock-out strains. S.N. and F.B. helped with software and hardware development, and with tuning beads secretion assay protocols. F.B. and G.B. supervised the study. S.S.-C. and G.B. wrote the manuscript with input from all authors.

## Competing interests

The authors declare no competing interests.
