## [Peer Review File · Nature Communications]

Reviewers' Comments:

Reviewer #1:

Remarks to the Author:

In this paper Sosa-Carrillo and colleagues present a highly detailed study of protein secretory mechanisms in yeast, as well as mechanisms for their optimisation. The study is interdisciplinary, and impressive in many regards; it brings together synthetic biological development of a novel multiple-output state reporter system, a highly integrated and effective robotic experimental apparatus allowing collection of rich data sets, as well as mathematical models and biological hypotheses that elegantly tie together their many observations. In addition to advancing our fundamental understanding of factors that determine protein production and secretion rates in yeast, the study also extends this fundamental advancement to applications related to real-time control of bioprocesses. Overall the paper is methodologically cutting-edge and yields novel insights of broad interest, and these are generally presented in a clear and detailed manner which will undoubtedly benefit future readers.

Following the above, in my opinion it is a well-designed, highly original, and impressive study that is likely to be of great interest both to fundamental researchers in yeast and synthetic biology, as well as applied researchers in bioprocess engineering and manufacturing. I am therefore glad to strongly support the work's publication in Nature Communications.

Below are some minor suggestions for potential improvements to the manuscript; none of these should require collection of additional data, and mostly they focus on improving presentation, discussion, and analysis of the results. I hope the authors may consider some of these points in their resubmission.

* In the introduction a great deal of explanation is given for the yeast secretory mechanism, touching on points including secretion, adaptation, stress, UPR, etc. There is a lot of terminology to get on top of in order to understand how these factors interlink. Might it be possible to include a simple cartoon in Figure 1 (or a new earlier figure) that conceptually sketches how the relevant processes interrelate? For example along the lines of that in the top left of Fig4. This may make it easier to understand some sections - e.g. Page 5 penultimate paragraph where it is stated that despite XylC placing a higher load on secretion pathways (compared to mNeon) the cell is able to "adapt" better to compensate - what exactly is going on here?

* Page 2 - Last sentence - can you say more about this accessory strain approach and its motivation? I appreciate broadly why it helps, but did not think the purpose was clearly justified. In particular, you state in the main text that it is there "to ensure a minimal media flow through the reactors", however given it is present in 1:10 ratio how much does this impact the actual minimal media flow? I can imagine it has a big impact if the growth of the producer strain is less than 1/10th of that of the accessory strain - but in such a case wouldn't the accessory strain very quickly dilute out the producer strain of interest? In this same direction, in Supplementary section 2.2 it is stated another purpose for the accessory is to help you MEASURE growth rates, which again seems sensible, but not exactly what is stated in the main text. In the supplementary it motivates this by comparing two methods for growth rate estimation - using quantification of input pump rate to estimate growth rate or the relative growth of the accessory strain. Could you comment on how these might compare with a "dithering" approach in which Optical Density is allowed to follow a zig-zag pattern, and growth rate extracted from the rate of rise between successive dilution events? (e.g. see DOI: 10.1371/journal.pone.0181923). Finally on this part; for Figure S2.2 could you add labels to each line in the plots - currently it is unclear what they indicate.

* Page 3 - last sentence - might you expand more on what is meant by "long time scales" here, or perhaps in the methods/discussion later? What was the longest time you got one of the feedback control experiments to run for; what are the limiting factors in its longevity?

* Figure 1c - This might be clarified slightly to show how signals are passed through the system. It currently shows EL222 directly activating pLight, which seems sensible. However, here there is also a single red arrow linking the POI and the pStress promoter; I appreciate that this is the direction/sign of the interaction, however I understand that this interaction is actually mediated by cell stress/UPR. Could the figure therefore be adjusted to look something like "POI -> Stress -> pStress", so that it is clear to the reader that the POI isn't itself directly activating transcription in some way?

* Page 9 - Potential typo; should "not accumulators were observed..." read "no accumulators were observed..."?

* Page 11 - This section has very interesting discussion of the adaptation method, helping us to understand whether cells in the accumulator state are actively transitioning/recovering back to normal state, or if they are just being diluted out due to lower growth. The test done is to knock out the UPR mechanism, but it was unclear to me whether this would tell the "whole" story regarding which of the above mechanisms were responsible. In my opinion an important distinction here is between "cell adaptation" and "population adaptation", and in this case it seems to me that you are looking at the latter as you have a turbidostat setup whereby what happens in one member of the population impacts another (i.e. if one cell type grows quickly it increases the relative rate at which others are diluted). With this in mind, would it be possible to include some additional simple mathematical analysis here to quantitatively calculate the impact of reduced growth in the accumulator cells on their dilution? In particular, you already know the average culture growth rate and fraction of accumulator cells (from data in Figure 3). From this you could calculate approximately how long it should take the system to adapt from this two-phenotype state if adaptation was solely due to dilution due to reduced growth of accumulators. For example, you might assume accumulators have approximately zero growth (compared to non accumulators which you assume grow as in other experiments) and from this back-calculate how long it should take a given accumulator population to dilute out. Alternatively, you could use some of the UPR knockout data to get an approximate growth rate for the stalled cells. Either way, you would be able to ballpark estimate how long it would take the population as a whole to adapt (given its continual growth) under this alternative mechanism, further supporting your analysis on page 11.

* Page 16 - Here the benefits of the feedback control mechanism are discussed, and indeed the results seem very supportive. However, one thing not discussed is how much the presence of dynamic feedback control benefits the system over just setting an open-loop controller using the characterisation data for varying induction strengths generated previously. It seems based on the top right plot in Figure 6 that one could obtain a ~50% gain in production (i.e. the height of the top red dot) just by setting the open loop set point correctly - admittedly this is not as good as the 70% from the closed system, but some discussion would be worthwhile highlighting when the benefits/limitations of each approach may be relevant. I.e. one imagines the closed loop system would be better in the face of any perturbations to the population from changes in external environment, media quality etc. Again, I am not suggesting you go back to the lab and test this as it is beyond the scope of the study, but it may be worthwhile to mention in the context of your broader discussion.

Harrison Steel

Reviewer #2:

Remarks to the Author:

In this article, the authors demonstrate an interesting application of optogenetic feedback control to maximize protein secretion in budding yeast. The authors control the production of several different proteins (of increasing complexity as assessed by post-translational modifications) under the EL222 promoter so that protein production is light inducible. Strains are grown in a turbidostat setup, and automatically sampled into a flow cytometer for assessment of accumulated protein (tagged with mNeonGreen) and unfolded protein response induction (mScarlet). Secreted protein is measured using accumulation on magnetic beads through a FLAG tag and subsequent washing and assessment of bead brightness in a flow cytometer. An accessory strain which expresses

mScarlet under the EL222 promoter (but which is distinguishable from the production strain by two copies of cerulean) is used to keep track of light intensity/induction (based on mScarlet induction) as well as growth rate (by comparing the production strain growth rate to the accessory strain growth rate which is assumed to be constant). The authors show that induction of proteins (and especially more complex proteins) causes an accumulator phenotype in a subpopulation of cells. By using bang-bang control to keep populations of cells near the level of stress where this subpopulation emerges, the authors are able to show some optimization of protein secretion. The paper contains an impressive amount of data for protein secretion, UPR, and secretion across a variety of light-induction levels and for several different proteins. I also appreciate that the experiments are quite technically challenging, particularly quantifying secreted protein, and the authors have come up with several interesting, albeit slightly complex, solutions including the connection of the turbidostats to flow cytometry and measurement of secreted protein using magnetic beads. Overall this paper is an important contribution to the field of cybergenetics and demonstrates an application for real-time control of cellular populations. In order to be appropriate for publication, I feel that the authors should address at least the major comments below. In particular, the paper would benefit from a rewrite with a careful eye to clarity and also to being clear about what claims can and cannot be made based on the experiments in the paper. It is interesting that this work and the recent preprint from Benisch took a similar approach to this problem (with important differences) and are not completely aligned in their findings. I don't think that this detracts from the importance of this work in any way, but rather both are important contributions (and in fact, comparing and contrasting the two is likely to be informative).

Major Comments:

- The abstract or introduction should say what strain of yeast the authors are working with. Related to this, was there a compelling reason for picking BY4741 which is usually not an industrially relevant strain?
- Can the authors clarify why it is necessary to measure stress levels and why a priori this is what they thought to assess in trying to optimize production? Would it be possible to optimize simply using light levels and protein output alone? How does the speed of feedback from stress levels compare to the feedback from measuring accumulators or protein output?
- What happens to the ratio of accumulator to production strain, and how continuous can this culture system be? At what point does the accessory strain overtake the producing strain? In a real production situation, I assume that there wouldn't be an accessory strain, but here it is critical to make important measurements.
- Are the only light dynamics explored modulating duty cycle on a 30- or 45-minute period? It would be interesting to see how modulating intensity vs period vs duty cycle of the light pulses changes the induction level and stress response of the cells. Was there a reason for picking the 30- or 45-minute periods? In general, the choices made in the optogenetic system and light induction pattern are not explained/elaborated well in the paper. Why EL222? Why 45 minute periods? Why the light intensities used in the reactors (and indeed, what is the range of light intensities used in the reactors?) Are there any controls showing that light doesn't affect cell growth or exacerbate accumulator phenotypes/stress (for instance due to repeatedly exciting GFP?)
- The origin of secretion burnout in a subpopulation of cells is not well explained. What causes the heterogeneity and why is this subpopulation "burning out"?
- The modeling is done on the non-burnout population, and the modeling is used to indirectly show what kind of feedback is present. One question is why the modelling on the non-burnout population is relevant for understanding what is happening with the burnout population. Another is whether or not there is a more direct way to measure what feedback is happening to adjust protein secretion? It would be more convincing, for example, to measure protein degradation directly.
- Some of the design choices are not clear, in that it isn't clear why more circuitous routes needed to be taken. For example, why couldn't light intensity from the LEDs be measured and calibrated as is done in other optogenetic studies, rather than needing an accessory strain to back-out the induction strength after the fact? (See for example Biotechniques or MethodsX paper from the McClean lab, Tabor lab LPA paper, optoPlate protocol from the Bugaj lab, etc). Similarly, why couldn't growth rates be inferred from the supply pumps (p3 "When working in turbidostat mode, the OD is kept constant and one can in principle infer the growth rate from the influx rate of the supply pumps. These estimates were not very precise")? Is this a fundamental limitation of the

system, or an engineering problem that could be solved? Both the lack of measurement on the light intensity and the growth rate required the use of the accessory strain, which is an interesting solution, but one with additional drawbacks (for example, the accessory strain should be able to outcompete the production strain and sets a limit on the time these experiments can be run).

- Was constancy of the accessory strain growth rate ever directly tested and is this a reasonable assumption?

- Supplemental p8: More details are needed on the label free proteomics. How was protein measured to get the iBAQ values? More details on the methodology are either needed here or in the Methods section.

- P11: "This observation demonstrates that UPR-mediated adaptation is the principal, if not unique, mechanism of adaptation for cells experiencing secretion burnout." This isn't a direct measurement. It is possible that the accumulator cells are reaching the same phenotype as the hac1 deletions through a different mechanism. If you overexpress HAC1 does the accumulator phenotype go away? I don't know that it is necessary for this paper to determine exactly why the accumulator phenotype is occurring, since the focus of this paper is on controlling protein secretion and the accumulator phenotype is a useful way to gauge when to do that, but I do think you need to be careful in terms of what we actually know about the accumulators based on the experiments in this paper.

- P14: Why is ktrf decreasing for all of the experiments as a function of induction level? Does it make sense that this rate would be regulated? In general, how overfit is the model and how much can we trust the fits to these kinetic parameters? Indeed, that is why it would be useful to measure some of these directly if understanding the feedback mechanism is important.

- P14 "Firstly, one identifies the optimal level of stress" Is it clear what the optimal level of stress is? The authors do optimize protein secretion by using bang-bang control, although how optimal the solution isn't clear. Indeed, in Figure 6 many of the blue dots overlap with the red dots (characterization experiments) except for one blue star which seems maximal. So does this really demonstrate that control is optimizing protein production, and if so, is this really the optimal control strategy? (More below) Also why is keeping protein stress below a certain threshold optimal? Is it because the accumulators don't wash out, or why does this give a significant gain in secreted protein?

Comments on Figures:

- Fig 1e: It isn't clear what the light and dark blue denotes in these figures.

- Fig 2: Why do the iPOI and UPR traces in time go out to 15ish hours (estimating from the x-axis) but the iPOI data vs induction level is taken at 24 hours? Can the whole timecourse be shown on the left (or shown in the supplemental to show that a steady-state is reached that continues to 24 hours?)

- Fig 3: Ought to have a legend for light intensity. Description states that intensity of the blue lines corresponds to light intensity, but it is also mentioned on pg 5, p 1 that the LEDs aren't calibrated and that the induction level is determined by the accessory strain. Might also not be a bad idea to split the figure into more lettered subsections. Also, should the caption read "overflowed" instead of "overflown"? Or maybe "overflowing"?

- Fig 5 description: states that intensity of the blue color in dots corresponds to the induction levels. I would like more clarification as to whether the dots correspond to light intensity or induction levels. Does it vary from figure to figure? How are these calculated?

- Fig 6: The top right graph in 6b shows a blue star, indicating that experiment is the one shown in the top left. The description states that all blue dots are real-time control experiments. So what makes the blue star special? Did it have the best control parameters? Did you just cherry pick it? The graph shows it as being 70% higher than constant light induction, but how much better is it than just optimizing a constant light induction regime? Is there really a significant benefit to the real-time control aspect here? The 45 minute sampling and 45 minute induction period do add a significant delay to the feedback control. (This is related to comments above)

- Fig 6: This whole figure is using the scFv-secreting cells? This should be mentioned in the description. I would also be interested to see how effective this technique is when applied to some of the other strains that don't demonstrate such a dramatic sweet spot. Was this tested?

- Figure S2.1b: Is slope really what you want here? Would some measure of the correlation between mNeonGreen and mScarlet (which must be very high looking at the plot) be more convincing? Does the numerical value of the slope matter much?

- Figure S2.2: Details are lacking here. What are the different colored lines (gray vs light blue vs

dark blue)? It would be easier to understand if each subpanel had a caption (a,b,c,d). Are spike-in cells the same as the accessory strain...please change the axis label to match the language in the caption. How is the growth rate calculated, i.e., what is the moving window size? I could not find these details in the supplemental material or in the methods.

- Figure S6.2: More details on how the relative decrease in growth rate is calculated.

Minor Comments:

- In the abstract, the 70% claim should be qualified by specifying that it's only for the secretion levels of some proteins (scFv).

- Pg2 "and reporting for their UPR secretory stress". The wording here is a bit kludgy and it isn't clear that you mean that these strains have a reporter for the UPR secretory stress.

- Pg3 p1: First mention of *Saccharomyces cerevisiae* should have *Saccharomyces* spelled out.

- Pg3 p1: Should a reference for the FLAG purification tag be included?

- Pg3 "we co-cultured an accessory strain having a constant growth rate (Supplementary Note 2)"

It isn't clear at this point what the accessory strain is for? I.e. why does the accessory strain ensure a minimal flow of media? I realize that this is explained in Supplementary note 2, but maybe just a quick clarification here/succinct description of how the accessory strain is used and why it is necessary. Details are obviously fine in the Supplemental.

- P6 "To do so, we took a volume of the cell culture directly from the reactors at different levels of light induction and after 24 hours of induction. Further information on beads measurements is provided in the thesis manuscript of Sebastian Sosa-Carrillo7." This information would be better included in the paper.

- P7 "The two samples from cell cultures are shown after washing most of the cells by the protocol explained in materials and methods of the main text" How long do these washing steps take, and does this affect the real-time feedback or does the real-time feedback not take into account the secreted protein concentrations?

- Pg11 p2: Change "any additional demand" to "additional demand." Induction of a protein under pTDH3-VP16-EL222 is not negligible.

- Pg12 p1: Is it actually easier to track appearance of accumulators than to quantify secreted protein levels? Adding an extra genetic construct to track UPR levels means you could use the same construct for multiple strains and wouldn't need to modify your protein with a tag for easy quantification.

- Pg14 p1: "previous findings" makes it sound like you are referring to previous works, rather than this work.

- Pg20 p1: Should the vendor-recommended procedure for bead equilibration be reproduced here?

- Pg9 p1 should be non-accumulators instead of not accumulators

- Pg11 p2: Should be "stops growing" instead of "stop growing"

- Pg13 p1: should read "Increasing the demand further" instead of "Increasing further the demand"

- Pg14 p1: I don't love "apparition of accumulators." It sounds better to say "appearance of accumulators."

- Pg17 p1 Should be "some processes of interest"

- Pg17 p3 Should be "Transformations were"

- Supplementary Note 2 indicates that the accessory strain has two copies of cerulean and that is why it is distinguishable from the production strain, but the strain table doesn't make clear that yLB44 has two copies. In general,

- Is HAC1 deficient the same as the *hac1* deletion? (I'm guessing yes, based on the strain table). In which case, please keep the nomenclature consistent throughout the text. *Hac1* deficient would imply a hypoactive mutant or reduction in expression, rather than a deletion.

- Is it appropriate to call the control scheme used in this paper bang-bang control, and if so, could that be stated? Alternatively, what is the name of the control scheme used?

- P. 15 "Apparition of accumulator" Is this meant to say "appearance of accumulator cells"?

- P. 20 For Prd, why is the rate of induction and degradation the same (K_{prd})? And should these two rates even have the same units, what are the units of Ind?

- Supplementary Note 8: More useful than the ReacSight code would be a flow diagram, or other representation of the algorithm that doesn't require understanding the code syntax (although keeping the code in the supplemental is fine/useful).

Reviewer #3:

Remarks to the Author:

The authors make some statements that should be modulated:

Results section on "Maximising protein production using real-time control and optimal stress levels and Fig 6a, right figure: "protein production (induction levels)", fig 4 caption.

In some parts of this section/figures, it gives the impression that the the transcriptional activity (induction level) of the promoter can be actually tuned by light intensity. But in fact, the induction by light appears to be an on/off switch, and the parameter that actually can be varied is the duration of the light stimulation. So, in a way, the protein synthesis rate is always the same, cannot be adjusted to the secretion rate, which would be more advantageous from the control point of view. This should be clarified and the text modified accordingly to avoid conceptual misunderstanding

Real-time control means a very short time delay between measurement and control action. What is the actual time lapse between the detection of increase in UPR level and switching on light stimulation?

I do not fully understand the concept of "% of sampling period" used to define the increase in light stimulation time. Please explain (and add explanation in materials and methods cultivation set up).

The authors seem to assume that the transcriptional activity of the minimal promoter CYC180 is constant over the range of growth rates of the essays. Is that really the case? Given that the producing strains do not keep a constant growth rate in the turbidostat, it would be important to assess this potential effect.

Regarding the heterogeneity of cell populations that the authors observe in terms of protein production capacity. Could this be related to the phase of the cell cycle they are? Some literature on this topic could be worth to consider when discussing the results of this study.

Discussion page 17: the authors point at potential industrial applicability of their control strategy of UPR levels based on a light-regulated promoter for protein production. However, it is difficult to imagine how light-control of protein production could be implemented in a large scale fermenter (usually made of stainless steel, operated in fed-batch, i.e. high cell densities where light scattering may be an issue). The authors should clarify what do they mean by industrial applications.

Finally, they also refer to the challenge of identifying stress responsive promoters that can be appropriately used to close the regulation loop. There are indeed some publications regarding this topic, testing a range of different UPR-responsive promoters such as KAR2, ERO1, etc. The authors should consider these previous studies to enrich this point of the discussion.

Minor points:

Methods:

Page 18: the name of the organism (*S. cerevisiae*) should be stated, not just a strain number. In addition, please provide the genotype of this strain and a reference or source, either in the text or in table S1.2. In table S1.2, other parental strains are also mentioned. For consistency, they should be also mentioned in page 18. Otherwise, do not provide any strain number in page 18 and simply refer to the S1 table with the full information.

Page 18: what type of small bioreactor was used? is it a commercially available bioreactor type (e.g. AMBR)? if not, please provide a full description of the equipment, eg. what sort of stirring and aeration system is used etc, or refer to an article where the cultivation system is described in detail.

Materials and methods-Analytical methods: How is UPR measured? that is, which sensor (UPR-responsive promoter) do you use? I could not find the information in M&M section and in Table S1 is not specified. This is an important aspect of the methodology that should be fully explained,

including the choice of UPR-responsive promoter, as there are several choices (KAR2, ERO1, etc), and they all respond differently.

We thank all reviewers for their positive comments on our work and for their suggestions to improve the article.

Below we provide detailed comments to their remarks.

Reviewer #1

In this paper Sosa-Carrillo and colleagues present a highly detailed study of protein secretory mechanisms in yeast, as well as mechanisms for their optimisation. The study is interdisciplinary, and impressive in many regards; it brings together synthetic biological development of a novel multiple-output state reporter system, a highly integrated and effective robotic experimental apparatus allowing collection of rich data sets, as well as mathematical models and biological hypotheses that elegant tie together their many observations. In addition to advancing our fundamental understanding of factors that determine protein production and secretion rates in yeast, the study also extends this fundamental advancement to applications related to real-time control of bioprocesses. Overall the paper is methodologically cutting-edge and yields novel insights of broad interest, and these are generally presented in a clear and detailed manner which will undoubtedly benefit future readers.

Following the above, in my opinion it is a well-designed, highly original, and impressive study that is likely to be of great interest both to fundamental researchers in yeast and synthetic biology, as well as applied researchers in bioprocess engineering and manufacturing. I am therefore glad to strongly support the work's publication in Nature Communications.

We thank the reviewer for his positive appreciation of our work.

Below are some minor suggestions for potential improvements to the manuscript; none of these should require collection of additional data, and mostly they focus on improving presentation, discussion, and analysis of the results. I hope the authors may consider some of these points in their resubmission.

R1.1: In the introduction a great deal of explanation is given for the yeast secretory mechanism, touching on points including secretion, adaptation, stress, UPR, etc. There is a lot of terminology to get on top of in order to understand how these factors interlink. Might it be possible to include a simple cartoon in Figure 1 (or a new earlier figure) that conceptually sketches how the relevant processes interrelate? For example along the lines of that in the top left of Fig4. This may make it easier to understand some sections - e.g. Page 5 penultimate paragraph where it is stated that despite XylC placing a higher load on secretion pathways (compared to mNeon) the cell is able to "adapt" better to compensate – what exactly is going on here?

Protein secretion relies on a rather complex set of biological processes, interconnected with the entire physiology of the cell. We feel that the explanations provided on protein secretion in the second paragraph of the introduction already correspond to the minimal amount of information to properly introduce the topic in the context of current knowledge. Following the advice of the reviewer, we included a cartoon representation of this process in Figure 1 (panel e, coming from Figure 2a). This gives us an opportunity to restate in even simpler terms the biology of the process that we study:

If the external demand is such that the production of the heterologous protein exceeds the secretory capacity of the cell, proteins might accumulate, thereby triggering stress adaptation responses that lead to an increase of the secretory capacity and/or to an increase in protein degradation capacity.

This representation cannot be placed even earlier (e.g., as a panel a of Figure 1), because it integrates a simplified representation of the biology together with the specificities of our experimental setup (it focuses on what we can control or measure).

R1.2: Page 2 - Last sentence - can you say more about this accessory strain approach and its motivation? I appreciate broadly why it helps, but did not think the purpose was clearly justified. In particular, you state in the main text that it is there "to ensure a minimal media flow through the reactors", however given it is present in 1:10 ratio how much does this impact the actual minimal media flow? I can imagine it has a big impact if the growth of the producer strain is less than 1/10th of that of the accessory strain - but in such a case wouldn't the accessory strain very quickly dilute out the producer strain of interest? In this same direction, in Supplementary section 2.2 it is stated another purpose for the accessory is to help you MEASURE growth rates, which again seems sensible, but not exactly what is stated in the main text. In the supplementary it motivates this by comparing two methods for growth rate estimation - using quantification of input pump rate to estimate growth rate or the relative growth of the accessory strain. Could you comment on how these might compare with a "dithering" approach in which Optical Density is allowed to follow a zig-zag pattern, and growth rate extracted from the rate of rise between successive dilution events? (e.g. see DOI: 10.1371/journal.pone.0181923). Finally on this part; for Figure S2.2 could you add labels to each line in the plots - currently it is unclear what they indicate.

The accessory strain serves three purposes. It is used to guarantee a minimal flow of media through the reactors operated in turbidostat mode since its growth is not affected by secretory burden, to quantify the actual induction demand in the strain of interest, and to assess its growth rate. This was stated in the caption of Figure 1, but not consistently in the main text. We have improved this in the text and in the header of Supplementary note 2. Its role to maintain media flow was essential when characterizing *HAC1* knock-out strains for example (Figure 4). In such severe cases, we do observe that the accessory strain dilutes out the strain of interest. This might lead to issues in cytometry measurements.

Regarding growth rate estimates, one can indeed obtain good estimates using the proposed "dithering" (zig zag) approach. Actually, this is how we estimated the growth rate of the accessory strain (our "reference" growth rate). However, by definition, the cell density fluctuates in these experiments. Because we were using the accessory strain for other purposes, we simply took advantage of its presence.

Some information was indeed missing in Figure S2.2. We extended the caption to provide the missing information.

3: Page 3 - last sentence - might you expand more on what is meant by "long time scales" here, or perhaps in the methods/discussion later? What was the longest time you got one of the feedback control experiments to run for; what are the limiting factors in its longevity?

Experiments in this paper typically last close to two days in the bioreactor: 16 hours without cytometry ("preculture"), 4 hours of cytometry measurements prior to light stimulation, and 24 hours of cytometry measurements with light stimulation. Longer experiments have been published in our *ReacSight* paper (doi: 10.1038/s41467-022-31033-9, Figure 3d, 1 day "preculture" + 5 days

of cytometry measurements) and in our yeast differentiation paper (doi: 10.1038/s41467-021-26129-7, Figure 3b, 1 day “preculture” + 5 days of cytometry measurements).

R1.4: Figure 1c - This might be clarified slightly to show how signals are passed through the system. It currently shows EL222 directly activating pLight, which seems sensible. However, here there is also a single red arrow linking the POI and the pStress promoter; I appreciate that this is the direction/sign of the interaction, however I understand that this interaction is actually mediated by cell stress/UPR. Could the figure therefore be adjusted to look something like "POI -> Stress -> pStress", so that it is clear to the reader that the POI isn't itself directly activating transcription in some way?

True. We clarified this issue as proposed.

R1.5: Page 9 - Potential typo; should "not accumulators were observed..." read "no accumulators were observed..."?

Indeed. This was a typo.

R1.6: Page 11 - This section has very interesting discussion of the adaptation method, helping us to understand whether cells in the accumulator state are actively transitioning/recovering back to normal state, or if they are just being diluted out due to lower growth. The test done is to knock out the UPR mechanism, but it was unclear to me whether this would tell the "whole" story regarding which of the above mechanisms were responsible. In my opinion an important distinction here is between "cell adaptation" and "population adaptation", and in this case it seems to me that you are looking at the latter as you have a turbidostat setup whereby what happens in one member of the population impacts another (i.e. if one cell type grows quickly it increases the relative rate at which others are diluted). With this in mind, would it be possible to include some additional simple mathematical analysis here to quantitatively calculate the impact of reduced growth in the accumulator cells on their dilution? In particular, you already know the average culture growth rate and fraction of accumulator cells (from data in Figure 3). From this you could calculate approximately how long it should take the system to adapt from this two-phenotype state if adaptation was solely due to dilution due to reduced growth of accumulators. For example, you might assume accumulators have approximately zero growth (compared to non accumulators which you assume grow as in other experiments) and from this back-calculate how long it should take a given accumulator population to dilute out. Alternatively, you could use some of the UPR knockout data to get an approximate growth rate for the stalled cells. Either way, you would be able to ball-park estimate how long it would take the population as a whole to adapt (given its continual growth) under this alternative mechanism, further supporting your analysis on page 11.

We thank the reviewer for this suggestion, and we have performed the proposed analysis. Assuming that our simple model assumptions hold, we found that not all scFv accumulator cells are washed away, meaning that some cells are adapting fast enough to avoid being outcompeted by non-accumulators and accessory cells. The picture is somewhat different for many accumulators. In this case, it seems that these cells do not adapt and are all eventually washed away. This analysis is detailed in a novel section of the Supplementary note 6 (section 6.3).

R1.7: Page 16 - Here the benefits of the feedback control mechanism are discussed, and indeed the results seem very supportive. However, one thing not discussed is how much the presence of dynamic feedback control benefits the system over just setting an open-loop controller using the characterisation data for varying induction strengths generated previously. It seems based on the top right plot in Figure 6 that one could obtain a ~50% gain in production (i.e. the height of the top red dot) just by setting the open loop set point correctly - admittedly this is not as good as the 70% from

the closed system, but some discussion would be worthwhile highlighting when the benefits/limitations of each approach may be relevant. I.e. one imagines the closed loop system would be better in the face of any perturbations to the population from changes in external environment, media quality etc. Again, I am not suggesting you go back to the lab and test this as it is beyond the scope of the study, but it may be worthwhile to mention in the context of your broader discussion.

We agree with the reviewer that, based on the data we have, it is possible that a carefully chosen open loop control strategy can perform equally well than the closed loop control experiment we show. Setting up this open loop strategy amounts to finding via trials and errors the induction level that triggers a stress that is close to the target stress level used in our closed loop control experiment. The use of a closed loop control strategy is an alternative to this trial-and-error approach for open loop control. Close loop control strategies might therefore be more time efficient. Moreover, the relation between induction and secretion stress is protein-dependent and media-dependent. We anticipate, although we do not demonstrate it, that closed loop control strategies offer more versatile and robust solutions than those based on open loop control. We added this discussion in the main text.

Harrison Steel

Reviewer #2

In this article, the authors demonstrate an interesting application of optogenetic feedback control to maximize protein secretion in budding yeast. The authors control the production of several different proteins (of increasing complexity as assessed by post-translational modifications) under the EL222 promoter so that protein production is light inducible. Strains are grown in a turbidostat setup, and automatically sampled into a flow cytometer for assessment of accumulated protein (tagged with mNeonGreen) and unfolded protein response induction (mScarlet). Secreted protein is measured using accumulation on magnetic beads through a FLAG tag and subsequent washing and assessment of bead brightness in a flow cytometer. An accessory strain which expresses mScarlet under the EL222 promoter (but which is distinguishable from the production strain by two copies of cerulean) is used to keep track of light intensity/induction (based on mScarlet induction) as well as growth rate (by comparing the production strain growth rate to the accessory strain growth rate which is assumed to be constant). The authors show that induction of proteins (and especially more complex proteins) causes an accumulator phenotype in a subpopulation of cells. By using bang-bang control to keep populations of cells near the level of stress where this subpopulation emerges, the authors are able to show some optimization of protein secretion.

The paper contains an impressive amount of data for protein secretion, UPR, and secretion across a variety of light-induction levels and for several different proteins. I also appreciate that the experiments are quite technically challenging, particularly quantifying secreted protein, and the authors have come up with several interesting, albeit slightly complex, solutions including the connection of the turbidostats to flow cytometry and measurement of secreted protein using magnetic beads. Overall this paper is an important contribution to the field of cybergenetics and demonstrates an application for real-time control of cellular populations. In order to be appropriate for publication, I feel that the authors should address at least the major comments below. In particular, the paper would benefit from a rewrite with a careful eye to clarity and also to being clear about what claims can and cannot be made based on the experiments in the paper. It is interesting

that this work and the recent preprint from Benisch took a similar approach to this problem (with important differences) and are not completely aligned in their findings. I don't think that this detracts from the importance of this work in any way, but rather both are important contributions (and in fact, comparing and contrasting the two is likely to be informative).

We thank the reviewer for their positive appreciation of our work.

Major Comments:

R2.1 - The abstract or introduction should say what strain of yeast the authors are working with. Related to this, was there a compelling reason for picking BY4741 which is usually not an industrially relevant strain?

We clarified in the introduction that we are working with *Saccharomyces cerevisiae*. This information was indeed missing. In this article, our central question is to better understand the relation between heterologous protein secretion and (the preservation of) cell physiology. The choice of the specific strain BY4741 is motivated by the fact that we have a slightly more pronounced focus on the fundamental / systems biology side of the question rather than on the applied / bioproduction side of the question. We are also making a heavy use of the MoClo Yeast ToolKit developed by Dueber and colleagues (doi: 10.1021/sb500366v) for genetic constructions, and this tool kit has been primarily characterized with the BY4741 strain.

R2.2 - Can the authors clarify why it is necessary to measure stress levels and why a priori this is what they thought to assess in trying to optimize production? Would it be possible to optimize simply using light levels and protein output alone? How does the speed of feedback from stress levels compare to the feedback from measuring accumulators or protein output?

We aim at providing solutions that are rather generic, ideally that are independent of the specific heterologous protein secreted. Establishing a control strategy based on the level of secreted protein in the media is challenging since this would require real-time quantification of secreted protein levels. Establishing a control strategy based on the level of internal protein in cells is easy to implement with our approach but would require that the protein is necessarily fused to a fluorescent reporter. This might not be desirable for some applications. In contrast, establishing a control strategy based on the stress level of the cells is in principle doable for any strain of interest (even heavily engineered ones) since it only necessitates the addition of a fluorescent reporter for secretion stress.

From our measurements, it appears that the iPOI signal increases slightly before the UPR stress signal. Therefore, a real-time control strategy based on the former can in principle be more efficient than a similar control strategy based on the latter. Yet, as mentioned above, we think that control strategies based on stress levels have the potential to be rather generic and should therefore be preferred. We do not have data on the dynamics of protein accumulation in the media. Therefore, we cannot discuss the possible performance of a controller based on secreted protein levels.

R2.3 - What happens to the ratio of accumulator to production strain, and how continuous can this culture system be? At what point does the accessory strain overtake the producing strain? In a real production situation, I assume that there wouldn't be an accessory strain, but here it is critical to make important measurements.

Our real-time control strategy precisely aims at avoiding the imposition of a too strong burden on cells. Therefore, producer cells should not (and do not) exhibit growth defects in these control

experiments. So, a co-culture with accessory strains would not be an issue (the initial ratio would be maintained). In addition, given that, in control experiments light intensity is provided based on the stress level of producer cells, accessory strains are then of no use. And indeed, our control experiments are performed in absence of accessory cells.

In summary, the accessory strain is very useful in characterization experiments but is dispensable in control experiments.

R2.4 - Are the only light dynamics explored modulating duty cycle on a 30- or 45-minute period? It would be interesting to see how modulating intensity vs period vs duty cycle of the light pulses changes the induction level and stress response of the cells. Was there a reason for picking the 30- or 45-minute periods? In general, the choices made in the optogenetic system and light induction pattern are not explained/elaborated well in the paper. Why EL222? Why 45 minute periods? Why the light intensities used in the reactors (and indeed, what is the range of light intensities used in the reactors?) Are there any controls showing that light doesn't affect cell growth or exacerbate accumulator phenotypes/stress (for instance due to repeatedly exciting GFP?)

The optogenetic system we use has been extensively well characterized (see e.g., Refs 27 and 31, and our works, Ref 22 and 41). This system has many advantages. One can notably mention that it is fast and reversible, it functions as a homodimer, and it does not necessitate the addition of cofactors.

The motivation for using duty cycles rather than intensity to modulate gene expression has been documented in Benzinger and Khammash (Ref 31). These authors observed that in comparison to intensity modulation, pulsatile light stimulations lead to a reduced cell-to-cell variability in gene expression.

We use a 30-minute period for light stimulations since Benzinger and Khammash showed that such pulsatile light stimulations give rise to a protein production that is an almost perfectly linear function of the duty cycle (Fig 2c in Benzinger and Khammash). We use a 45-minute period for sampling times. This corresponds approximately to the time needed to run cytometry measurements for the 8 samples coming from our 8 bioreactors, wash the cytometer plate with the robot and process the data. Note that for real-time control experiments, light stimulation periods have been aligned with sampling times, to ease the implementation.

In Figure 3d and in Supplementary Figure 11 of the ReacSight paper (Bertaux, Sosa, *et al.*, *Nat Comm*, 2022), we show in a 5-day long experiment, that a continuous light stimulation of strains expressing a non-secreted or a secreted version of mNeonGreen leads to no noticeable changes in gene expression over at least 48 hours.

R2.5 - The origin of secretion burnout in a subpopulation of cells is not well explained. What causes the heterogeneity and why is this subpopulation "burning out"?

Because of the stochasticity of gene expression and possibly other stochastic processes, cells are heterogeneous in their capacities to produce proteins, to translocate them in the ER, to secrete them and to perform short-term adaptation to secretion stress. Based on the observations reported in Fig 3a and Supplementary note 6, we deduced that proteins accumulate in the secretory pathway of cells that produce proteins at levels exceeding their protein trafficking capacity and that do not adapt rapidly enough. This leads to an accumulator phenotype, and is associated in these cells with a high level of stress and a drop in growth rate (secretion burn-out). Given the results of Fig 4, it appears that rapid adaptation capabilities play an important role in

preventing the appearance of the accumulator phenotype shortly after induction. Yet, we do not know the mechanistic origin(s) of this phenotype. Is it determined by one key factor or is it the result of a combination of multiple factors? This question is very interesting but has remained unanswered so far. We mention explicitly in our discussion that “the molecular mechanisms remain to be clarified”, such that it is clear that our results do not provide explanations for this.

R2.6 - The modeling is done on the non-burnout population, and the modeling is used to indirectly show what kind of feedback is present. One question is why the modelling on the non-burnout population is relevant for understanding what is happening with the burnout population. Another is whether or not there is a more direct way to measure what feedback is happening to adjust protein secretion? It would be more convincing, for example, to measure protein degradation directly.

“why the modelling on the non-burnout population is relevant for understanding what is happening with the burnout population”: the modeling is done on the non-burnout population and the conclusions are drawn for the non-burnout cells. In the paragraph of Fig 5, we say explicitly that “we focused on understanding the adaptation response of the non-accumulator cells”. In our summary statement we refer to non-burnout cells (“In summary, our quantitative analysis reveals that at the induction levels where accumulators appear, cells have reached their maximal secretory adaptation capabilities. Increasing further the demand leads to increasing protein degradation in the non-accumulator cells”), and for completeness just add a reference to already known properties of the burnout cells (“...or transiently experiencing a secretion burnout in accumulator cells”).

We agree with the reviewer that measuring protein degradation activities in cells should give a more direct demonstration that protein degradation increases when induction exceeds the accumulator-appearance threshold. However, proteins might be degraded via different pathways and constructing genetic strains that quantitatively report on their degradation activities appeared challenging to us. We therefore relied on a more indirect approach using models and parameter inference.

R2.7: - Some of the design choices are not clear, in that it isn't clear why more circuitous routes needed to be taken. For example, why couldn't light intensity from the LEDs be measured and calibrated as is done in other optogenetic studies, rather than needing an accessory strain to back-out the induction strength after the fact? (See for example Biotechniques or MethodsX paper from the McClean lab, Tabor lab LPA paper, optoPlate protocol from the Bugaj lab, etc). Similarly, why couldn't growth rates be inferred from the supply pumps (p3 “When working in turbidostat mode, the OD is kept constant and one can in principle infer the growth rate from the influx rate of the supply pumps. These estimates were not very precise”)? Is this a fundamental limitation of the system, or an engineering problem that could be solved? Both the lack of measurement on the light intensity and the growth rate required the use of the accessory strain, which is an interesting solution, but one with additional drawbacks (for example, the accessory strain should be able to outcompete the production strain and sets a limit on the time these experiments can be run).

As stated in our answer to Remark 1.2 of Reviewer #1, the accessory strain serves three purposes. It is used to guarantee a minimal flow of media through the reactors since its growth is not affected by secretory burden, to quantify the actual induction demand in the strain of interest, and to assess its growth rate. These 3 different purposes were not clearly stated in the main text. The coculture was notably important to ensure media renewal when a significant fraction of the cells were growth-arrested as it was the case in *HAC1*-deletion strains (Figure 4). It was also convenient to properly estimate the light intensity perceived by the cells. Indeed, small day-to-day variations

have been observed with our platform. This could originate from variations in LED intensities, or more likely, from small differences in the placement of the sampling or air supply lines within the vessel. Therefore, the use of the accessory strain is convenient on several counts to provide quantitative information. But it is not essential to our approach. It was not used in real-time control experiments for example.

R2.8: Was constancy of the accessory strain growth rate ever directly tested and is this a reasonable assumption?

We use bioreactors in turbidostat mode with relatively low target ODs ($OD^* = 0.5$) to have an excellent control on growth conditions. As shown in Supplementary note 2, a coculture of the accessory strain and of an amylase-secreting strain is stable over at least 24 hours in absence of light, that is, in absence of a burden for the latter strain. No deviation of the initial 1:10 ratio is observed in such cases.

R2.9: Supplemental p8: More details are needed on the label free proteomics. How was protein measured to get the iBAQ values? More details on the methodology are either needed here or in the Methods section.

We added in Supplementary note 4 the detailed protocol used to obtain our proteomics results.

R2.10: P11: "This observation demonstrates that UPR-mediated adaptation is the principal, if not unique, mechanism of adaptation for cells experiencing secretion burnout." This isn't a direct measurement. It is possible that the accumulator cells are reaching the same phenotype as the *hac1* deletions through a different mechanism. If you overexpress *HAC1* does the accumulator phenotype go away? I don't know that it is necessary for this paper to determine exactly why the accumulator phenotype is occurring, since the focus of this paper is on controlling protein secretion and the accumulator phenotype is a useful way to gauge when to do that, but I do think you need to be careful in terms of what we actually know about the accumulators based on the experiments in this paper.

We agree that there might be two distinct ways to reach the accumulator phenotype in normal and in *HAC1* knockout strains. However, our claim is on the need for *Hac1* to adapt. Strains with *Hac1* do adapt, and strains without *Hac1* do not adapt. If there would be a *Hac1*-independent adaptation mechanism, strains without *Hac1* could adapt. This is not what we observe. We therefore conclude that *Hac1* is needed for cell adaptation.

Maybe, we are missing something here to completely understand the remark of the reviewer. We remain open to modulating our statements if need be. As rightly mentioned by the reviewer, the focus on our paper is not on this aspect.

R2.11: P14: Why is *k_{trf}* decreasing for all of the experiments as a function of induction level? Does it make sense that this rate would be regulated? In general, how overfit is the model and how much can we trust the fits to these kinetic parameters? Indeed, that is why it would be useful to measure some of these directly if understanding the feedback mechanism is important.

The decrease of the estimated value of *k_{trf}* with increasing induction levels is indeed not explained in the text. This should correspond to the saturation of the transcription, translation, or ER translocation capacities. Because the intracellular level of non-secreted mNeonGreen correlates linearly with the induction strength, it is not likely that transcription and translation saturate with increasing induction levels. We therefore hypothesize that translocation in the ER is becoming a limiting factor. However, we can not prove this.

We fully agree that a direct measurement of some of these rates would be beneficial for our study. Unfortunately, we have not found a satisfying solution to measure them in live cells. Indeed, protein translocation in the ER and subsequent posttranslational modifications, on the one side, and degradation of non-mature proteins, on the other side, are both complex, multi-stage processes.

We used a careful approach for parameter fitting and use simple reasoning on the structural identifiability of parameters (K_{prd}), prior information from calibration experiments (K_{prd} and K_{dil}), and additional measurements on secreted protein levels (K_{units}) to fully constrain the remaining 3 parameters we estimate (K_{trf} , K_{sec} , K_{deg}) based on time series data on iPOI and measured secreted protein concentrations.

R2.12 - P14 “Firstly, one identifies the optimal level of stress” Is it clear what the optimal level of stress is? The authors do optimize protein secretion by using bang-bang control, although how optimal the solution isn’t clear. Indeed, in Figure 6 many of the blue dots overlap with the red dots (characterization experiments) except for one blue star which seems maximal. So does this really demonstrate that control is optimizing protein production, and if so, is this really the optimal control strategy? (More below) Also why is keeping protein stress below a certain threshold optimal? Is it because the accumulators don’t wash out, or why does this give a significant gain in secreted protein?

The sentence “Firstly, one identifies the optimal level of stress, that is the UPR stress level that is associated with the apparition of accumulators.” has to be understood in its context. The previous sentence reads “The previous findings can be used to propose a real-time control strategy to maximize protein bioproduction”. If one assumes that the previous findings hold, then the optimal level of stress is well defined. It is when accumulators appear, that is, when secretion rates are high and degradation rates are still low. That being said, we do not have the guarantee that this induction level is indeed the optimal one, as noted by the reviewer. To introduce this distinction, the sentence has been rephrased as “Firstly, one identifies what should be the optimal level of stress, that is the UPR stress level that is associated with the apparition of accumulators.”

When one obtains high levels of secreted proteins in control experiments, is it thanks to the control actions themselves or thanks to keeping cells at an appropriate stress level? This remark has also been raised by the first reviewer and we refer to our answer to Remark 1.7. We added a discussion of this point in the main text.

“why is keeping protein stress below a certain threshold optimal?” Assuming that the findings of Fig 5 hold, then by keeping protein stress below the given threshold, we have that secretion rates are high, degradation rates are still low, and accumulators are present in marginal numbers. This is therefore a very advantageous situation.

Comments on Figures:

R2.13: Fig 1e: It isn’t clear what the light and dark blue denotes in these figures.

True. In the caption of the figure, we added that “the intensity of the blue color corresponds to the intensity of the light stimulation received by cells in the bioreactor, as reported by the co-cultured accessory strain. In our context, this corresponds to the protein production demand.”

R2.14: Fig 2: Why do the iPOI and UPR traces in time go out to 15ish hours (estimating from the x-axis) but the iPOI data vs induction level is taken at 24 hours? Can the whole timecourse be shown

on the left (or shown in the supplemental to show that a steady-state is reached that continues to 24 hours?)

The full data has not been displayed in the main text because of space limitation in the figure. Based on the reviewer's recommendation we added the complete time courses in Supplementary note 5.

R2.15: Fig 3: Ought to have a legend for light intensity. Description states that intensity of the blue lines corresponds to light intensity, but it is also mentioned on pg 5, p 1 that the LEDs aren't calibrated and that the induction level is determined by the accessory strain. Might also not be a bad idea to split the figure into more lettered subsections. Also, should the caption read "overflowed" instead of "overflown"? Or maybe "overflowing"?

The legend for induction levels was indeed missing and has been added. Note that, we do not report light intensities, but directly induction levels as reported by the accessory strain. The motivation for this is now better explained in Supplementary note 2.

The typo has been corrected.

R2.16: Fig 5 description: states that intensity of the blue color in dots corresponds to the induction levels. I would like more clarification as to whether the dots correspond to light intensity or induction levels. Does it vary from figure to figure? How are these calculated?

We realized that the notion of normalized induction levels that we use extensively has not been properly introduced. In the Method section, we clarified that "Normalized induction levels are defined as the red fluorescence intensity of the co-cultured accessory strain divided by the maximal red fluorescence intensity of the accessory strain found across all experiments shown in this study. Therefore, normalized induction levels can be directly compared across experiments."

R2.17: Fig 6: The top right graph in 6b shows a blue star, indicating that experiment is the one shown in the top left. The description states that all blue dots are real-time control experiments. So what makes the blue star special? Did it have the best control parameters? Did you just cherry pick it? The graph shows it as being 70% higher than constant light induction, but how much better is it than just optimizing a constant light induction regime? Is there really a significant benefit to the real-time control aspect here? The 45 minute sampling and 45 minute induction period do add a significant delay to the feedback control. (This is related to comments above)

The different real-time control experiments differ by their target stress levels. In this sense, the fact that production is maximal when the target is set to the stress level at which accumulators appear confirms that this stress level is an optimal one for production (see answer to remark 2.12).

We agree with the reviewer that, based on the data we have, it is possible that a carefully chosen open loop control strategy (a constant light induction regime) can perform equally well than the closed loop control experiment we show. This question was also raised by Reviewer #1 and we refer to our answer to Remark 1.7. In summary, we anticipate that closed loop control strategies offer more versatile and robust solutions than those based on open loop control. We added a discussion of this question in the main text.

It is true that the fact that we perform control actions only every 45 minutes creates a significant delay to steer the system. This can notably explain the presence of overshoot in the experiment

shown in Fig 6b. Yet, our main message is not on the potential of real-time control by itself but on the importance of keeping cells at their production sweet spots.

R2.18: Fig 6: This whole figure is using the scFv-secreting cells? This should be mentioned in the description. I would also be interested to see how effective this technique is when applied to some of the other strains that don't demonstrate such a dramatic sweet spot. Was this tested?

Indeed. This was mentioned in the main text but not in the caption of the figure. This omission has been corrected.

We focused on the scFv-secreting cells for our control experiments since the highest secretion level is obtained for these cells at intermediate inductions levels. This is not a feature shared by the other proteins we tested here and therefore, we would not anticipate that great benefits would be obtained for the other proteins we characterized in this study. So, we have not tested them.

R2.19: Figure S2.1b: Is slope really what you want here? Would some measure of the correlation between mNeonGreen and mScarlet (which must be very high looking at the plot) be more convincing? Does the numerical value of the slope matter much?

We agree with the reviewer that the slope of the regression line was not very informative in this case. We replaced this information by the value of the correlation coefficient that makes more sense in this context.

R2.20: Figure S2.2: Details are lacking here. What are the different colored lines (gray vs light blue vs dark blue)? It would be easier to understand if each subpanel had a caption (a,b,c,d). Are spike-in cells the same as the accessory strain...please change the axis label to match the language in the caption. How is the growth rate calculated, i.e., what is the moving window size? I could not find these details in the supplemental material or in the methods.

The figure has been updated and missing information has been added in the caption. The terminology of "spike-in strain" has been replaced by the one of "accessory strain". This occurrence has escaped our attention. The method used for growth rate estimations has been clarified as well.

R2.21: Figure S6.2: More details on how the relative decrease in growth rate is calculated.

We clarified this point and added a note on how the slope of the linear regression curve can be interpreted in terms of accumulator cell growth rate.

Minor Comments:

R2.22: In the abstract, the 70% claim should be qualified by specifying that it's only for the secretion levels of some proteins (scFv).

We added this precision in the abstract.

R2.23: Pg2 "and reporting for their UPR secretory stress". The wording here is a bit kludgy and it isn't clear that you mean that these strains have a reporter for the UPR secretory stress.

We rephrased the sentence to address this issue.

R2.24: Pg3 p1: First mention of *Saccharomyces cerevisiae* should have *Saccharomyces* spelled out.

The first mention of *Saccharomyces cerevisiae* is now in the introduction and is not abbreviated.

R2.25: Pg3 p1: Should a reference for the FLAG purification tag be included?

We added a reference for the FLAG tag.

R2.26: Pg3 “we co-cultured an accessory strain having a constant growth rate (Supplementary Note 2)” It isn’t clear at this point what the accessory strain is for? I.e. why does the accessory strain ensure a minimal flow of media? I realize that this is explained in Supplementary note 2, but maybe just a quick clarification here/succinct description of how the accessory strain is used and why it is necessary. Details are obviously fine in the Supplemental.

The presentation of the role of the accessory strain was incomplete in the main text. We expanded our explanations to clarify this point.

R2.27: P6 “To do so, we took a volume of the cell culture directly from the reactors at different levels of light induction and after 24 hours of induction. Further information on beads measurements is provided in the thesis manuscript of Sebastian Sosa-Carrillo⁷.” This information would be better included in the paper.

We are preparing a publication that documents the automation of this protocol using the experimental platform we use. This protocol and significant improvements thereof will be presented in detail in this coming publication. In the meantime, we simply refer to the publicly-available thesis of Sebastian Sosa Carrillo.

R2.28: P7 “The two samples from cell cultures are shown after washing most of the cells by the protocol explained in materials and methods of the main text” How long do these washing steps take, and does this affect the real-time feedback or does the real-time feedback not take into account the secreted protein concentrations?

The real-time control routine uses exclusively the UPR stress levels of the sampled cells. Protein concentration measurements are not used.

For information, as said in the method section for Secretion measurements, the washing is done after incubation of the cell culture with the beads during 1 hour at room temperature. Then, the samples are washed three times in 200 μ L of TBS buffer using a magnetic rack to retain beads. Finally, before measurements the beads are collected and resuspended again in 200 μ L of TBS buffer. The total duration of the process, including incubation, but not the time within the cytometer, is about 1 hour and 10 minutes. Again, protein measurements are not part of the real-time feedback loop.

R2.29: Pg11 p2: Change “any additional demand” to “additional demand.” Induction of a protein under pTDH3-VP16-EL222 is not negligible.

We rephrased this sentence.

R2.30: Pg12 p1: Is it actually easier to track appearance of accumulators than to quantify secreted protein levels? Adding an extra genetic construct to track UPR levels means you could use the same construct for multiple strains and wouldn’t need to modify your protein with a tag for easy quantification.

Yes, we believe that it is easier to track the appearance of accumulators than to quantify secreted protein levels. The justifications provided by the reviewer are convincing.

R2.31: Pg14 p1: “previous findings” makes it sound like you are referring to previous works, rather than this work.

This has been addressed.

R2.32: Pg20 p1: Should the vendor-recommended procedure for bead equilibration be reproduced here?

The beads equilibration procedure consists essentially in washing the magnetic beads four times in TBS buffer to remove the glycerol in which beads are stored for long-term conservation. The protocol proposed by the vendor provides a number of technical information that are generally useful but that appear unnecessarily detailed here.

R2.33: Pg9 p1 should be non-accumulators instead of not accumulators

This has been addressed.

R2.34: Pg11 p2: Should be “stops growing” instead of “stop growing”

This has been addressed.

R2.35: Pg13 p1: should read “Increasing the demand further” instead of “Increasing further the demand”

This has been addressed.

R2.36: Pg14 p1: I don't love “apparition of accumulators.” It sounds better to say “appearance of accumulators.”

OK. We have modified this term.

R2.37: Pg17 p1 Should be “some processes of interest”

This has been addressed.

R2.38: Pg17 p3 Should be “Transformations were”

This has been addressed.

R2.39: Supplemental Note 2 indicates that the accessory strain has two copies of cerulean and that is why it is distinguishable from the production strain, but the strain table doesn't make clear that yLB44 has two copies. In general,

This was a mistake. The accessory strain (yIB337) has two copies of *mCerulean*. We corrected this in Table S1.2.

R2.40: Is HAC1 deficient the same as the hac1 deletion? (I'm guessing yes, based on the strain table). In which case, please keep the nomenclature consistent throughout the text. Hac1 deficient would imply a hypoactive mutant or reduction in expression, rather than a deletion.

When referring to genetic modifications, we use the “*HAC1* knock-out strain” terminology. When referring to functional aspects, we use the “adaptation-deficient strain” and “Hac1-deficient strain” terminology. We hope that in this context the use of “deficient” is appropriate.

R2.41: Is it appropriate to call the control scheme used in this paper bang-bang control, and if so, could that be stated? Alternatively, what is the name of the control scheme used?

It is true that our control scheme is inspired by a bang-bang controller. Yet, in bang-bang controllers, the input can only take two values (u_{\min} et u_{\max}). So, it would not be totally

appropriate to use this terminology for our controller. It could be called a piecewise-constant, discrete-time controller. This would not provide significant additional information, however.

R2.42: P. 15 “Apparition of accumulator” Is this meant to say “appearance of accumulator cells”?

Yes. This has been modified.

R2.43: P. 20 For Prd, why is the rate of induction and degradation the same (Kprd?) And should these two rates even have the same units, what are the units of Ind?

As mentioned in the main text, the protein production rate parameter and the mRNA degradation rate parameter cannot be estimated independently based on the experimental data we have. To deal with this non-identifiability issue, we set them equal. This amounts to renormalizing the Prd values. They should in principle be given two distinct names, however slightly abusing notations, we give them a same name since they have the same value. Ind stands for the normalized induction level and is therefore unitless. Therefore, the dimensions of the protein production rate parameter are concentration over time, and the dimension of the mRNA degradation rate parameter is the inverse of a time.

R2.44: Supplementary Note 8: More useful than the ReacSight code would be a flow diagram, or other representation of the algorithm that doesn't require understanding the code syntax (although keeping the code in the supplemental is fine/useful).

An article on ReacSight has been published very recently. In that manuscript, we provided many details on the tool and its use. We notably provided examples showing real time control experiments. Therefore, rather than providing a simple scheme of the functioning of the software, we prefer to recommend to the interested reader to read the detailed article. We therefore added in the Supplementary note that “Details on the software and on the experimental platform can be found in Bertaux, Sosa, et al., Nat Commun, 2022 (Ref 22 of the main text)”.

Reviewer #3

The authors make some statements that should be modulated:

R3.1: Results section on "Maximising protein production using real-time control and optimal stress levels and Fig 6a, right figure: "protein production (induction levels)", fig 4 caption.

In some parts of this section/figures, it gives the impression that the the transcriptional activity (induction level) of the promoter can be actually tuned by light intensity. But in fact, the induction by light appears to be an on/off switch, and the parameter that actually can be varied is the duration of the light stimulation. So, in a way, the protein synthesis rate is always the same, cannot be adjusted to the secretion rate, which would be more advantageous from the control point of view. This should be clarified and the text modified accordingly to avoid conceptual misunderstanding

It is true that our induction strategy relies on switching light on or off. We follow a duty cycle encoding of the light stimulation rather than a more intuitive intensity encoding. To obtain a 50% induction level, rather than continuously applying a light stimulation with a 50% intensity, we apply full light half of the time in a periodic fashion. For this optogenetic system, Benzinger and Khammash have proven that the latter strategy is advantageous on several counts: such pulsatile light stimulations give rise to protein levels that follow an almost perfectly linear function of the

duty cycle (Fig 2c in Benzinger and Khammash, Nat Comm, 2018). Moreover, cell to cell variability is reduced in comparison to the intensity-modulation based strategy.

When the reviewer argues that “in a way, the protein synthesis rate is always the same”, they are right if this is understood at any given time point: it is either maximal or null (if one neglects cell to cell variability at least). However, when averaged over a period, the protein synthesis rate takes graded values.

We clarified this in the main text when introducing our light stimulation method:

To achieve different levels of production demand using the EL222 optogenetic expression system, we varied the duration of the light exposure within a constant time period of 30 minutes. That is, we adopt here the duty-cycle encoding of light stimulations as documented in Benzinger and Khammash³¹

R3.2: Real-time control means a very short time delay between measurement and control action. What is the actual time lapse between the detection of increase in UPR level and switching on light stimulation?

I do not fully understand the concept of "% of sampling period" used to define the increase in light stimulation time. Please explain (and add explanation in materials and methods cultivation set up).

A short delay between measurement and control action simplifies the implementation of control strategies. However, we note that in model-based control strategies this is not strictly needed if one can accurately predict the behavior of the system. For model-free approaches, the actuation delay should be small in comparison to the evolution time scale of the controlled process. Here, we have a 45-minute lag between sampling times and control actions, and a 2-hour cell generation time that influences protein dilution rates and hence protein levels. It is possible that using smaller actuation delays, the quality of our control experiment could be improved. However, this would imply that we decrease the number of experiments we run in parallel or that we add a second cytometer to the platform.

We clarified in the main text what we meant by “% of sampling period”

When UPR levels were below the reference level, the duration of light stimulation is increased by a fixed amount (5% of sampling period, that is, 2 minutes and 15 seconds given that the sampling time is 45 minutes).

R3.3 The authors seem to assume that the transcriptional activity of the minimal promoter CYC180 is constant over the range of growth rates of the essays. Is that really the case? Given that the producing strains do not keep a constant growth rate in the turbidostat, it would be important to assess this potential effect.

We agree with the reviewer that it is not reasonable to assume that transcriptional activity is unaltered in cells that exhibit decreased growth rates and severely altered physiology.

However, growth rate decays are only observed when accumulators are present. This is documented in Figure S6.1 and 6.2. Therefore, one can reasonably assume that only accumulators show a decreased growth rate. We developed a simple model-based analysis in a novel section of the Supplementary note 6 to investigate further this question.

Our quantitative analysis of protein production/translocation/secretion/degradation/dilution due to growth is therefore exclusively performed for the subpopulation on the non-accumulator cells

(Figure 5b). For this population, it seems reasonable to assume that transcription and translation do not vary significantly. Yet, we observe that the translocation rates K_{trf} decrease with increasing induction levels (Figure 5b). As discussed in the answer to Remark 2.11 of Reviewer #2, we believe that this is due to a decrease in ER translocation efficiency. Yet, given the data we have, we cannot rule out that the decrease of the K_{trf} rates correspond in fact to a decrease of the transcription or of the translation rates....

R3.4 Regarding the heterogeneity of cell populations that the authors observe in terms of protein production capacity. Could this be related to the phase of the cell cycle they are? Some literature on this topic could be worth to consider when discussing the results of this study.

There are indeed connections between cell cycle and secretion stress. Previous studies have shown that ER-associated stress impacts cell cycle and inhibits ER inheritance to the daughter cells, thus making them inviable (Jonas et al. 2018, Babour et al. 2010). This effect may exacerbate the accumulation of internal protein in the cells with overwhelmed secretory capacity, enhancing the accumulator phenotype due to lack of dilution. We added a reference to this information in the main text. The existence of a specific connection between the cell cycle phase and the appearance of accumulators is an interesting question that should probably be studied using a microscopy/microfluidic setup.

R3.5: Discussion page 17: the authors point at potential industrial applicability of their control strategy of UPR levels based on a light-regulated promoter for protein production. However, it is difficult to imagine how light-control of protein production could be implemented in a large scale fermenter (usually made of stainless steel, operated in fed-batch, i.e. high cell densities where light scattering may be an issue). The authors should clarify what do they mean by industrial applications.

Our proposed strategy is to tune down induction to prevent excessive stress. The detection of excessive stress is based on simple measurements of mean UPR stress levels in cells and can therefore be applied independently of a specific genetic background by the simple addition of a stress reporter gene. In particular, it is in principle compatible with and complementary to the extensive chassis-engineering optimization strategies found in industrial applications.

Even using standard chemical inducers, this criterion can be used to better control the induction phase. Inducers could be added gradually until reaching the target stress level. This might be less flexible and precise than using optogenetic control though. Whether optogenetic induction can be appropriate for industrial applications has been discussed in several recent reviews (e.g. Reshetnikov *et al.*, *Trends in Biotechnology*, 2022, Pouzet *et al.*, *Bioengineering*, 2020). Restating these discussions in the main text of our paper is probably not appropriate.

R3.6: Finally, they also refer to the challenge of identifying stress responsive promoters that can be appropriately used to close the regulation loop. There are indeed some publications regarding this topic, testing a range of different UPR-responsive promoters such as KAR2, ERO1, etc. The authors should consider these previous studies to enrich this point of the discussion.

There are many possible choices for secretory stress activated promoters, including those native from yeast as well as synthetic designs. From the control viewpoint, choosing an early responsive promoter should ease the implementation of a closed loop regulation. In most cases however, ER stress response promoters are triggered by the activation of Hac1. Because of its specific mode of action (the *HAC1*'s mRNA needs to be spliced and then translated), Hac1 regulated promoters might not be the best ones in terms of responsiveness. We are currently investigating this question using transcriptomics approaches. Moreover, we also added in the discussion a recent contribution

by Peng *et al* (ACS Synthetic Biology, 2021) on the characterization of novel biosensors for secretory stress.

Minor points:

Methods:

R3.7: Page 18: the name of the organism (*S. cerevisiae*) should be stated, not just a strain number. In addition, please provide the genotype of this strain and a reference or source, either in the text or in table S1.2. In table S1.2, other parental strains are also mentioned. For consistency, they should be also mentioned in page 18. Otherwise, do not provide any strain number in page 18 and simply refer to the S1 table with the full information.

We added the missing information in the Methods section and in the Supplementary note 1. No other parental strain has been used.

R3.8: Page 18: what type of small bioreactor was used? is it a commercially available bioreactor type (e.g. AMBR)? if not, please provide a full description of the equipment, eg. what sort of stirring and aeration system is used etc, or refer to an article where the cultivation system is described in detail.

We mentioned in the main text that we used “a dedicated multi-bioreactor platform with automated cytometry measurements and reactive optogenetic control of yeast in continuous cultures” and made a reference to our recent ReacSight paper (Bertaux, Sosa *et al.*, Nat Commun, 2022). This paper describes the platform in great detail. We added in the Methods section that “details on the bioreactors and the experimental platform can be found in [this reference]”.

R3.9: Materials and methods-Analytical methods: How is UPR measured? that is, which sensor (UPR-responsive promoter) do you use? I could not find the information in M&M section and in Table S1 is not specified. This is an important aspect of the methodology that should be fully explained, including the choice of UPR-responsive promoter, as there are several choices (KAR2, ERO1, etc), and they all respond differently.

We based the design of our UPR reporter on prior studies using a combination of a crippled *CYC1* promoter with four copies of the 22bp core sequence UPR1 from *S. cerevisiae* placed upstream of a red fluorescent reporter (Pincus *et al.* 2010, Merlsamer *et al.* 2008, and Cox *et al.* 1993). We clarified this aspect in the Methods section and added a reference to the Cox *et al.* (1993) paper when introducing the UPR stress reporter in the main text, in addition to the Pincus *et al.* (2010) paper.

Reviewers' Comments:

Reviewer #1:

Remarks to the Author:

Thank you for addressing all of my review comments. Congratulations on the improved article.

Reviewer #2:

Remarks to the Author:

The authors have addressed mine, and other reviewers', comments through revised text and some additional supplementary material. While I don't feel like all of my major comments from the previous review were addressed in the manuscript, but rather explained in the rebuttal (which is appreciated, but doesn't clarify the manuscript) I feel that the paper is acceptable for publication. A few minor points would improve the manuscript even further:

R2.18: Fig 6: The authors' response about why they only tested that one protein should be included in the main text.

Discussion: They added a section to the discussion talking about open vs closed loop control. I think the question that would be more interesting to answer is whether closed loop control optimizing stress level is more efficient than closed loop control optimizing protein output.

Reviewer #3:

Remarks to the Author:

The authors have addressed correctly my questions and comments, so I believe the manuscript is now suitable for publication.

We thank again all reviewers for their positive comments on our work and suggestions to improve the article.

Below we provide comments to their remarks.

Reviewer #1

Thank you for addressing all of my review comments. Congratulations on the improved article.

We thank the reviewer for his positive appreciation of our work and efforts to improve the manuscript.

Reviewer #2

The authors have addressed mine, and other reviewers', comments through revised text and some additional supplementary material. While I don't feel like all of my major comments from the previous review were addressed in the manuscript, but rather explained in the rebuttal (which is appreciated, but doesn't clarify the manuscript) I feel that the paper is acceptable for publication.

We thank the reviewer for their positive appreciation of our revised work.

A few minor points would improve the manuscript even further:

R2.18: Fig 6: The authors' response about why they only tested that one protein should be included in the main text.

We added explicitly in the discussion: "We focused on scFv-secreting cells for our control experiments since the highest secretion level is obtained for these cells at intermediate induction levels. Because this feature is not shared by the other proteins we tested in this work, we do not expect to obtain significant benefits for the optimization of the secretion of the other proteins."

Discussion: They added a section to the discussion talking about open vs closed loop control. I think the question that would be more interesting to answer is whether closed loop control optimizing stress level is more efficient than closed loop control optimizing protein output.

Tracking the levels of the secreted protein in real time would likely require fusing it to a reporter protein. For many applications, this is not desired. In contrast, adding a fluorescent protein that reports for secretion stress is generally easily doable. We clarified this aspect in the discussion: "Lastly, we stress that the regulation strategy we propose here, based on real-time measurements of secretion stress levels in cells, applies in principle to any protein to secrete and is complementary with classical chassis-engineering strategies. It only requires the addition to the cells of a secretion stress level reporter. No modification of the secreted protein is needed."

Reviewer #3

The authors have addressed correctly my questions and comments, so I believe the manuscript is now suitable for publication.

We thank the reviewer for their positive appreciation of our revised work.